# Selective consolidation of learning and memory via recall-gated plasticity

**Jack W Lindsey\*, Ashok Litwin-Kumar\***

Zuckerman Mind Brain Behavior Institute, Columbia University, New York, United States

**Abstract** In a variety of species and behavioral contexts, learning and memory formation recruits two neural systems, with initial plasticity in one system being consolidated into the other over time. Moreover, consolidation is known to be selective; that is, some experiences are more likely to be consolidated into long-term memory than others. Here, we propose and analyze a model that captures common computational principles underlying such phenomena. The key component of this model is a mechanism by which a long-term learning and memory system prioritizes the storage of synaptic changes that are consistent with prior updates to the short-term system. This mechanism, which we refer to as recall-gated consolidation, has the effect of shielding long-term memory from spurious synaptic changes, enabling it to focus on reliable signals in the environment. We describe neural circuit implementations of this model for different types of learning problems, including supervised learning, reinforcement learning, and autoassociative memory storage. These implementations involve synaptic plasticity rules modulated by factors such as prediction accuracy, decision confidence, or familiarity. We then develop an analytical theory of the learning and memory performance of the model, in comparison to alternatives relying only on synapse-local consolidation mechanisms. We find that recall-gated consolidation provides significant advantages, substantially amplifying the signal-to-noise ratio with which memories can be stored in noisy environments. We show that recall-gated consolidation gives rise to a number of phenomena that are present in behavioral learning paradigms, including spaced learning effects, task-dependent rates of consolidation, and differing neural representations in short- and long-term pathways.

**\*For correspondence:**
jackwlindsey@gmail.com (JWL);
a.litwin-kumar@columbia.edu
(AL-K)

**Competing interest:** The authors declare that no competing interests exist.

## eLife assessment

This **fundamental** work proposes a novel mechanism for memory consolidation where short-term memory provides a gating signal for memories to be consolidated into long-term storage. The work combines extensive analytical and numerical work applied to three different scenarios and provides a **convincing** analysis of the benefits of the proposed model, although some of the analyses are limited to the type of memory consolidation the authors consider (and don't consider), which limits the impact. The work will be of interest to neuroscientists and many other researchers interested in the mechanistic underpinnings of memory.

## Introduction

Systems that learn and remember confront a tradeoff between memory acquisition and retention. Plasticity enables learning but can corrupt previously stored information. Consolidation mechanisms, which stabilize or render more resilient certain plasticity events associated with memory formation, are key to navigating this tradeoff (*Kandel et al., 2014*). Consolidation may be mediated both by molecular dynamics at the synapse level (synaptic consolidation) and dynamics at the neural population level (systems consolidation).

**Figure 1.** Schematic of short- and long-term memory systems across species and brain areas. (**A**) In mice and other mammals, hippocampal memories are consolidated into neocortex. (**B**) Zebrafinch song learning initially depends on LMAN but later requires only HVC-to-RA synapses in the song motor pathway. (**C**) In the *Drosophila* mushroom body (inset), short- and long-term memories depend on dopamine-dependent plasticity in the *γ* and *α* lobes, respectively.

In this work, we present a model and theoretical analysis of selective systems consolidation, with a focus on understanding the computational advantages it offers in terms of long-term learning and memory storage. Several prior theoretical studies have studied synaptic consolidation models from this perspective, providing descriptions of how synaptic consolidation affects the strength and lifetime of memories (*Fusi et al., 2005*; *Lahiri and Ganguli, 2013*; *Benna and Fusi, 2016*). In these studies, synapses are modeled with multiple internal variables, operating at distinct timescales, which enable individual synapses to exist in more labile or more rigid ('consolidated') states. Such models can prolong memory lifetime and recapitulate certain memory-related phenomena, notably power-law forgetting curves. Moreover, this line of work has established theoretical limits on the memory retention capabilities of any such synaptic model, and shown that biologically realistic models can approximately achieve these limits (*Lahiri and Ganguli, 2013*; *Benna and Fusi, 2016*). These theoretical frameworks leave open the question of what computational benefit is provided by *systems* consolidation mechanisms that take place in a coordinated fashion across populations of neurons.

The term systems consolidation most often refers to the process by which memories stored in the hippocampus are transferred to the neocortex (*Figure 1A*; *Squire and Alvarez, 1995*; *Frankland and Bontempi, 2005*; *McClelland et al., 1995*; *McClelland and Goddard, 1996*). Prior work has described the hippocampus and the neocortex as 'complementary learning systems', emphasizing their distinct roles: the hippocampus stores information about individual experiences, and the neocortex extracts structure from many experiences (*McClelland et al., 1995*; *McClelland and Goddard, 1996*). Related phenomena also occur in other brain systems. In rodents, distinct pathways underlie the initial acquisition and long-term execution of some motor skills, with motor cortex apparently passing off responsibility to basal ganglia structures as learning progresses (*Kawai et al., 2015*; *Dhawale et al., 2021*). A similar consolidation process is observed during vocal learning in songbirds, where song learning is dependent on the region LMAN but song execution can, after multiple days of practice, become LMAN-independent and rely instead on the song motor pathway (*Figure 1B*; *Warren et al., 2011*). Some insects also display a form of systems consolidation. Olfactory learning experiments in fruit flies reveal that short-term memory (STM) and long-term memory (LTM) retrieval recruit distinct neurons within the mushroom body, and the short-term pathway is necessary for LTM formation (*Figure 1C*; *Cervantes-Sandoval et al., 2013*; *Dubnau and Chiang, 2013*).

The examples above are all characterized by two essential features: the presence of two systems involved in learning similar information and an asymmetric relationship between them, such that learning in one system (the 'LTM') is facilitated by the other (the 'STM'). Moreover, mounting evidence indicates that across all these systems, there exist mechanisms that selectively modulate or gate consolidation into long-term storage. In flies, for instance, recent work has shown that short-term olfactory memory recall gates LTM storage via a disinhibitory circuit, such that repeated stimulus-outcome pairings are consolidated into LTM but once-presented pairings are not (*Awata et al., 2019*). A recent study in songbirds indicates that the rate at which song learning is consolidated into the song motor pathway is modulated by performance quality (*Tachibana et al., 2022*). Finally, a large body of work has shown that propensity of hippocampal memories to be cortically consolidated is modulated

by a variety of factors including repetition, reliability, and novelty (*Terada et al., 2022*; *Gorriz et al., 2023*; *Jackson et al., 2006*; *Brodt et al., 2016*).

The ubiquity of the systems consolidation motif across species, neural circuits, and behaviors suggests that it offers broadly useful computational advantages that are complementary to those offered by synaptic mechanisms. In this work, we propose that the ability to selectively consolidate memories is the key computational feature that distinguishes systems from synaptic memory consolidation. To formalize this idea, we generalize prior theoretical studies studies by considering environments in which some memories should be prioritized more than others for long-term retention. We then introduce a model of selective systems consolidation and show that it can provide substantial performance advantages in such environments. In the model, synaptic updates are consolidated into LTM depending on their consistency with knowledge already stored in STM. We term this mechanism 'recall-gated consolidation'. This model is well-suited to prioritize storage of reliable patterns of synaptic updates which are consistently reinforced over time. We derive neural circuit implementations of this model for several tasks of interest. These involve plasticity rules modulated by globally broadcast factors such as prediction accuracy, confidence, and familiarity. We develop an analytical theory that describes the limits on learning performance achievable by synaptic consolidation mechanisms and shows that recall-gated systems consolidation exhibits qualitatively different and complementary benefits. Our theory depends on a quantitative treatment of environmental statistics, in particular the statistics with which similar events recur over time. Different model parameter choices suit different environmental statistics, and give rise to different learning behavior, including spaced training effects. The model makes predictions about the dependence of consolidation rate on the consistency of features in an environment, and the amount of time spent in it. It also predicts that STM benefits from employing sparser representations compared to LTM. A variety of experimental data support predictions of the model, which we review in the Discussion.

## Results

Following prior work (*Fusi et al., 2005*; *Benna and Fusi, 2016*), we consider a network of neurons connected by an ensemble of $N$ synapses whose weights are described by a vector $\mathbf{w} \in \mathbb{R}^N$ (our analysis also generalizes to synapses that store additional auxiliary information besides their weight; see Methods). For now, we are agnostic as to the structure of the network and its synaptic connections. The network's synapses are subject to a stream of patterns of candidate synaptic potentiation and depression events. We refer to such a pattern as a *memory*, defined by a vector $\mathbf{w}^* \in \mathbb{R}^N$. Synaptic weights are updated by memories according to a plasticity rule (see Methods). The plasticity rule always updates $\mathbf{w}$ to be more aligned with the memory vector $\mathbf{w}^*$; thus, the memory may be interpreted as a 'target' value for the synaptic weights. One simple example of a synaptic update rule is a 'binary switch' model, in which synapses can exist in two states (active or inactive), and candidate synaptic updates are binary (potentiation or depression). In this model, inactive synapses activate (resp. active synapses inactivate) in response to potentiation (resp. depression) events with some probability $p$. However, our systems consolidation model can be used with any underlying synaptic mechanisms, and we will consider a variety of synaptic plasticity rules as underlying substrates.

In our framework, the same memory can be encountered repeatedly over time, and we will refer to such repeated encounters as 'reinforcement' of a memory (not to be confused with reward-contingent notions of reinforcement). We distinguish memories by the *reliability*, or frequency, with which they are reinforced. The notion of reliability in our framework is meant to capture the idea that the structure of events in the world which drive synaptic plasticity is in some cases consistent over time, and in other cases inconsistent. For now, we focus on a simple environment model which captures this essential distinction, in which there are two kinds of memories: 'reliable' and 'unreliable'. Reliable memories are consistent patterns of synaptic updates that are reinforced regularly over time, whereas unreliable memories are spurious, randomly sampled patterns of synaptic updates. Concretely, in simulations, we assume that a given reliable memory is reinforced with independent probability $\lambda$ at each timestep, and otherwise a randomly sampled unreliable memory is encountered.

A useful measure of system performance is the memory *recall factor*, defined as the overlap $r = \mathbf{w} \cdot \mathbf{w}^*$ between a memory $\mathbf{w}^*$ and the current synaptic state $\mathbf{w}$. Specifically, we are interested in the signal-to-noise ratio (SNR) of the recall factor for reliable memories, (*Fusi et al., 2005*; *Benna and Fusi, 2016*), which normalizes the recall strength relative to the expected value of the fluctuations in

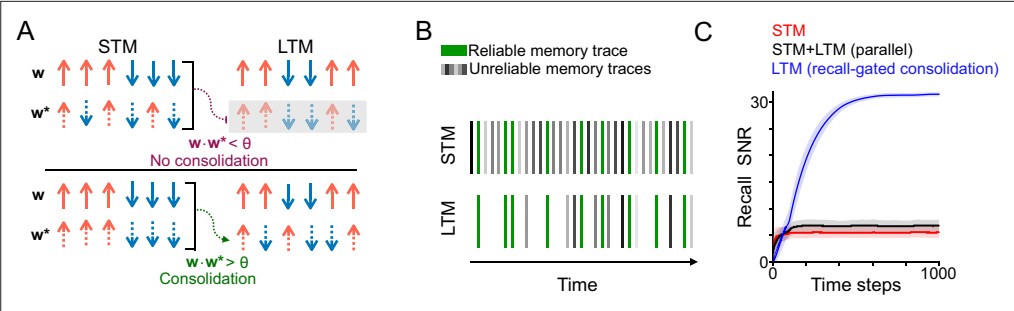

**Figure 2.** Recall-gated systems consolidation model. (**A**) Schematic of systems consolidation model. Top and bottom rows illustrate different examples, in which a memory is not consolidated or consolidated, respectively. Memories $\mathbf{w}^*$ correspond to patterns of candidate potentiation and depression events (dashed arrows) applied to a synaptic population with weights w (solid arrows). The synaptic population is divided into an STM (left) and LTM (right). Memories that provoke strong recall in the STM – that is, overlap strongly with the present synaptic state – enable plasticity (consolidation) in the LTM; otherwise plasticity in the LTM is gated (gray shaded rectangle). Note that the synaptic weights and the components of the memory corresponding to the LTM need not be linked to those of the STM (i.e. the patterns of arrows are different between the left and right columns). (**B**) Schematic of the environmental statistics. A reliable memory (green) arrives repeatedly with probability $\lambda$ at each time step, with randomly sampled 'unreliable' memories (gray) interspersed. The LTM is exposed to a filtered subset of consolidated memory traces with a higher proportion of reliable memories. (**C**) Simulation of recall performance of a single reliable memory with time as it is presented with probability $\lambda = 0.25$ at each time step, N=2000 synapses (1000 each in the STM and LTM). The STM and LTM learning rates (binary switching probabilities) are p=0.25 and p=0.05, respectively, and the synaptic state is initialized randomly, each synapse initially active with probability 0.5. In the recall-gated model, the gating threshold is set at $\theta = 2^{-3}$. Shaded regions indicate standard deviation across 1000 simulations.

The online version of this article includes the following figure supplement(s) for figure 2:

**Figure supplement 1.** Same as *Figure 2C*, but with multiple reliable memories simultaneously learned, each recurring equally often at a rate $\lambda_i = 0.01$ for all reliable memories $i$.

**Figure supplement 2.** SNR as a function of repetitions for single populations without consolidation, varying the parameter $k$ of the Weibull distribution governing interarrival times (and defining the learnable timescale in terms of the expected interarrival time).

recall factors for random memory vectors $\mathbf{w}^*_{\mathrm{rand}}$. We assume for simplicity that memories and weight vectors are mean-centered, so that the SNR may be written as follows:

$$\mathrm{SNR} = \frac{\mathbf{w} \cdot \mathbf{w}^*}{\sqrt{E_{\mathbf{w}^*_{\mathrm{rand}}}\left[\left(\mathbf{w} \cdot \mathbf{w}^*_{\mathrm{rand}}\right)^2\right]}}.$$

(1)

## Recall-gated systems consolidation

In our model (*Figure 2A*), we propose that the population of $N$ synapses is split into two subpopulations which we call the 'short-term memory' (STM) and 'long-term memory' (LTM). Upon every presentation of a memory $\mathbf{w}^*$, the STM recall $r_{\mathrm{STM}} = \mathbf{w}_{\mathrm{STM}} \cdot \mathbf{w}^*_{\mathrm{STM}}$ is computed. Learning in the LTM is modulated by a factor $g(r_{\mathrm{STM}})$. We refer to $g$ as the 'gating function'. For now we assume $g$ to be a simple threshold function, equal to 0 for $r_{\mathrm{STM}} < \theta$ and 1 for $r_{\mathrm{STM}} \geq \theta$, for some suitable threshold $\theta$. This means that consolidation occurs only when a memory is reinforced at a time when it can be recalled sufficiently strongly by the STM. Later we will consider different choices of the gating function $g$, which may be more appropriate depending on the statistics of memory recurrence in the environment.

We refer to this mechanism as *recall-gated consolidation*. Its function is to filter out unreliable memories, preventing them from affecting LTM synaptic weights. With an appropriately chosen gating function, reliable memories will pass through the gate at a higher rate than unreliable memories. Consequently, events that trigger plasticity in the LTM will consist of a higher proportion of reliable memories (*Figure 2B*), and hence the LTM will attain a higher SNR than the STM. The cost of this

gating is to incur some false negatives—reliable memory presentations that fail to update the LTM. However, some false negatives can be tolerated given that we expect reliable memories to recur multiple times, and information about these events is still present in the STM. As a proof of concept of the efficacy of recall-gated consolidation, we conducted a simulation in which memories correspond to random binary patterns and plasticity follows a binary switch rule (*Figure 2C*). This simulation implements an 'ideal observer' model in which we assume the system has direct access to the memory vector and can compute the recall factor exactly (realistic implementations are discussed below). Notably, recall-dependent consolidation results in reliable memory recall with a much higher SNR than an alternative model in which LTM weight updates proceed independently of STM recall.

## Neural circuit implementations of recall-gated consolidation

Our model requires a computation of recall strength, which we defined as the overlap between a memory and the current state of the synaptic population. From this definition, it is not clear how recall strength can be computed biologically. The mechanisms underlying computation of recall strength will depend on the task, network architecture, and learning rule giving rise to memory vectors. A simple example is the case of a population of input neurons connected to a single downstream output neuron, subject to a plasticity rule that potentiates synapses corresponding to active inputs. In this case, the recall strength quantity corresponds exactly to the total input received by the output neuron, which acts as a familiarity detector. Below, we give the corresponding recall factors for other learning and memory tasks: supervised learning, reinforcement learning, and unsupervised auto-associative memory, summarized in *Figure 3A*. The expressions for the recall factors are derived for each learning rule of interest in the Appendix. We emphasize that our use of the term 'recall' refers to the familiarity of *synaptic update patterns* (specifically, memory vectors), and does not necessarily correspond to familiarity of stimuli or other task variables. Thus, although we will continue to use the term 'recall factor', for a given task the recall factor quantity may have a different semantic interpretation, summarized in *Figure 3A*. Note also that we use the term 'learning rule' to refer to how the memory vector is constructed from task quantities, while 'plasticity rule' is reserved for the mechanism by which memory vectors update synaptic weights.

### Supervised learning

Suppose a population of neurons with activity $\mathbf{x}$ representing stimuli is connected via feedforward weights to a readout population with activity $\hat{\mathbf{y}} = \mathbf{W}\mathbf{x}$. The goal of the system is to predict ground-truth outputs $\mathbf{y}$. A simple choice of the form of the memory $\mathbf{W}^*$ (written with a capital letter since the synaptic weights are being interpreted as a matrix) that will train the system is $\mathbf{W}^* = \mathbf{y}\mathbf{x}^T$, corresponding to a associative Hebbian learning rule *Hebb, 1949*. The corresponding recall factor is $\mathbf{y} \cdot \hat{\mathbf{y}}$, corresponding to prediction accuracy (*Figure 3B*; see Appendix for derivation).

### Reinforcement learning

Suppose a population of neurons with activity $x$ representing an animal's state is connected to a population with activity $\boldsymbol{\pi} = \mathbf{W}\mathbf{x}$ which controls the action selection probabilities. Specifically, the log probability of selecting action $a$ is proportional to $\pi_a$ (see Methods). Following action selection, the animal receives a reward, the value of which depends on the state and chosen action. A simple approach to reinforcement learning is to use a memory vector arising from a 'three-factor' rule (*Joel et al., 2002*; *Frémaux and Gerstner, 2015*; *Lindsey et al., 2024*) $\mathbf{W}^* = \text{reward} \cdot \mathbf{a}\mathbf{x}^T$, where $\mathbf{a}$ is a vector with 1 in the index corresponding to the selected action and 0 elsewhere. This learning rule reinforces actions that lead to reward. For this model, the corresponding recall factor is $\text{reward} \cdot \pi_a$, a multiplicative combination of reward and the animal's confidence in its selected action, as measured by its a priori log likelihood of selecting that action (see Appendix for derivation). Intuitively, the recall factor will be high when a confidently chosen action leads to reward (*Figure 3C*).

### Unsupervised autoassociative memory

Suppose a population of neurons with activity $\mathbf{x}$ and recurrent weights $\mathbf{W}$ stores memories as attractors according to an autoassociative Hebbian rule, where memories correspond to $\mathbf{W}^* = \mathbf{x}\mathbf{x}^T$, similar to a Hopfield network *Hopfield, 1982*. In this case, the recall factor can be expressed as $\mathbf{x} \cdot (\mathbf{W}\mathbf{x})$, a comparison between stimulus input $\mathbf{x}$ and recurrent input $\mathbf{W}\mathbf{x}$ (see Appendix for derivation). Intuitively,

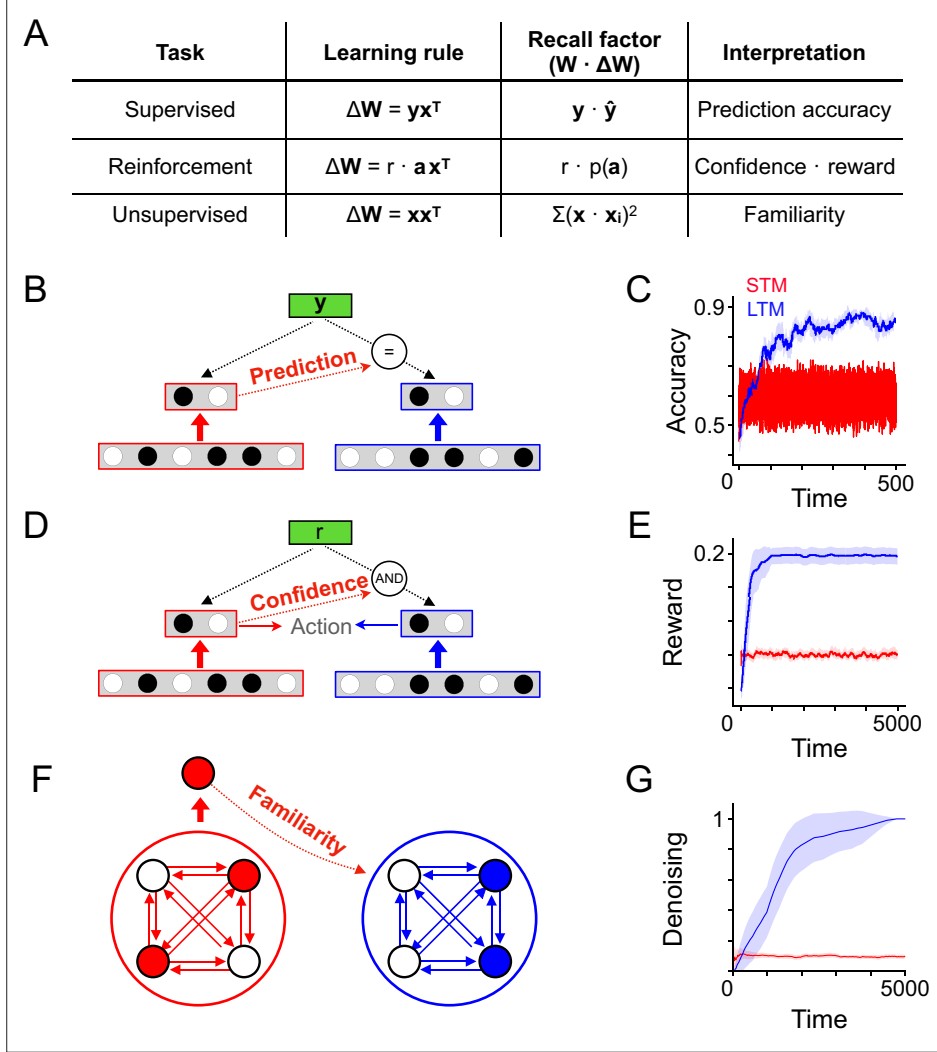

| Task | Learning rule | Recall factor (W · ΔW) | Interpretation |
|---|---|---|---|
| Supervised | $\Delta \mathbf{W} = \mathbf{y}\mathbf{x}^T$ | $\mathbf{y} \cdot \hat{\mathbf{y}}$ | Prediction accuracy |
| Reinforcement | $\Delta \mathbf{W} = r \cdot \mathbf{a}\mathbf{x}^T$ | $r \cdot p(\mathbf{a})$ | Confidence · reward |
| Unsupervised | $\Delta \mathbf{W} = \mathbf{x}\mathbf{x}^T$ | $\Sigma(\mathbf{x} \cdot \mathbf{x_i})^2$ | Familiarity |

**Figure 3.** Circuit architectures for recall-gated consolidation model. (**A**) Description of learning rules corresponding to different types of learning problems and corresponding expressions for the recall factor used in the recall-gated consolidation model. (**B**) Schematic indicating a possible implementation of the model in a supervised learning problem, where LTM plasticity is modulated by the consistency between STM predictions and ground-truth labels. (**C**) Like B, but for a reinforcement learning problem. LTM plasticity is gated by both STM action confidence and the presence of reward. (**D**) Like B and C, but for an autoassociative unsupervised learning problem. As above, $\mathbf{x}$ corresponds to neural activity and $\mathbf{W}$ to the network weights, which here are recurrent. LTM plasticity is gated by familiarity detection in the STM module. (**E**) Simulation of a binary classification problem, $N = 2000$, $\theta = 0.125$, $p = 0.1$. There are 20 total stimuli each associated with a random binary (±1) label and each appearing with probability $\lambda = 0.01$ at each timestep (otherwise a random stimulus is presented, with a random binary label). Plot shows the classification accuracy over time, given by the outputs of the STM and LTM of the consolidation model. Shaded region indicates standard deviation over 50 simulations. (**F**) Simulation of a reinforcement learning problem, $N = 2000$, $\theta = 0.125$, $p = 1.0$. There are five total stimuli, each appearing with probability $\lambda = 0.01$ at each timestep (otherwise a random stimulus is presented), and three possible actions. Each stimulus has a corresponding action that yields reward (the reward is randomly sampled for the random stimuli). The plot shows average reward per step over time, evaluated using the actions given by the STM or LTM (during learning, the STM action was always used). (**G**) Simulation of an autoassociative learning problem. $N = 4000$, $p = 1.0$. A single stimulus appears with probability $\lambda = 0.25$ at each timestep, and otherwise a random stimulus appears. Recall performance is evaluated by exposing the system to a noisy version of the reliable stimulus seen during training, allowing the recurrent dynamics of the network to run for 5 timesteps, and measuring the correlation of the final state of the network with the ground-truth pattern.

the recall factor measures the familiarity of the stimulus, as highly familiar stimuli will exhibit attractor behavior, making $\mathbf{x}$ and $\mathbf{Wx}$ highly correlated. Such a quantity could in principle be computed directly, for instance if separate dendritic compartments represent the feedforward inputs and the recurrent inputs, though such a mechanism is speculative. This quantity can also be approximated using a separate novelty readout trained alongside the recurrent weights, which is the implementation we use in our simulation. In this approach, a set of familiarity readout weights $\mathbf{u}$ receive the neural population activity as input, and outputs a scalar signal indicating the familiarity of that activity pattern. These familiarity readouts are updated according to their own corresponding memory vector $\mathbf{u}^* = \mathbf{x}$. The output of these weights, $\mathbf{u} \cdot \mathbf{x}$, estimates the familiarity of the activity pattern, and is used as the recall factor (see Appendix for more details on when this approximation is equal to the ideal recall factor).

To verify that the advantages of recall-gated consolidation illustrated in *Figure 2* apply in these tasks, we simulated the three architectures and learning rules described above (see Methods for simulation details). In each case, learning takes place online, with reliable task-relevant stimuli appearing a fraction $\lambda$ of the time, interspersed among randomly sampled unreliable stimuli. In the case of supervised and reinforcement learning tasks, unreliable stimuli are paired with random labels and random rewards, respectively. Reliable stimuli are associated with consistent labels or action-reward contingencies. We find that recall-gated consolidation provides significant benefits in each case, illustrating that the theoretical benefits of increased SNR in memory storage translate to improved performance on meaningful tasks (*Figure 3E, F and G*).

## An analytical theory of the recall of repeatedly reinforced memories

We now turn to analyzing the behavior of the recall-gated systems consolidation model more systematically, to understand the source of its computational benefits and characterize other predictions it makes. To do so, we developed an analytic theory of memory system performance, with and without recall-gated consolidation. To make the analysis tractable, our subsequent results assume an ideal observer model as in *Figure 2C*, where we assume the system has direct access to memory vectors and can compute the recall factor exactly. Importantly, our framework differs from prior work (*Fusi et al., 2005*; *Benna and Fusi, 2016*) in considering environments with intermittent repeated presentations of the same memory. We adopt several assumptions for analytical tractability. First, as in previous studies, we assume that inputs have been preprocessed so that the representations of different memories are random and uncorrelated with one another (*Gluck and Myers, 1993*; *Benna and Fusi, 2016*). We also assume, for now, that each memory consists of an equal number of positive and negative entries, although later we will relax this assumption. We are interested in tracking the SNR of recall for a given reliable memory. We emphasize that this quantity is an abstract measure of system performance reflecting the degree to which a specific set of synaptic changes (a memory trace) is retained in the system, and its interpretation varies according to the task in question (*Figure 3*).

The dynamics of memory storage depend strongly on the underlying synapse model and plasticity rule. Given a synaptic model, an important quantity to consider is its associated 'forgetting curve' $m(t)$, defined as the average SNR of recall for a memory $\mathbf{w}^*$ at $t$ timesteps following its first presentation, assuming a new randomly sampled memory has been presented at each timestep since. For example, the binary switch model with transition probability $p$ has an associated forgetting curve $m(t) \approx \sqrt{N}pe^{-pt}$ (*Fusi et al., 2005*). More sophisticated synapse models, such as the cascade model of *Fusi et al., 2005* and multivariable model of *Benna and Fusi, 2016* achieve power-law forgetting curves (see Methods). In the limit of large system size $N$ and under the assumption that memories are random, uncorrelated patterns, the forgetting curve is an exact description of the decay of recall strength.

Forgetting curves capture the behavior of a system in response to a single presentation of a memory, but we are concerned with the behavior of memory systems in response to multiple reinforcements of the same memory trace. Thus, another key quantity in our theory is the interarrival distribution $p(I)$, which describes the distribution of intervals between repeated presentations of the same memory, and its expected value $\tau = E[I]$, the average interval length. Our simplest case of interest is the case in which a given memory recurs according to a Poisson process; that is, it is reinforced with probability $\lambda$ at each timestep, independent of the history of recent reinforcements (as in the simulation in *Figure 2C*). This case corresponds to an exponential interarrival distribution $p(I) = \lambda e^{-\lambda x}$, with mean interarrival time $\tau = 1/\lambda$.

We now quantify the recall strength for a memory that has been reinforced $R$ times. For the synapse models we consider, this quantity can be approximated accurately (see Appendix) by summing the strengths of preceding forgetting curves, that is:

$$\text{SNR} \approx \sum_{i=1}^{R} m\left(t_i\right). \tag{2}$$

where $t_i$ is the time elapsed since the $i$th reinforcement of the memory. This quantity is a random variable whose value depends on the history of interarrival intervals of the memory, and the specific unreliable memories that have been stored in intervening timesteps. To more concisely characterize a system's memory performance, we introduce a new summary metric, the *learnable timescale* of the system. For a given target SNR value and allowable probability of error $\epsilon$, the learnable timescale $\tau_\beta^\epsilon$ is defined as the maximum interarrival timescale $\tau$ for which the SNR of recall will exceed $\beta$ with probability $1 - \epsilon$. We fix $\epsilon = 0.1$ throughout this work; this choice has no qualitative effect on our results. Learnable timescale captures the system's ability to reliably recall memories that are presented intermittently. We note that there exists a close relationship between learnable timescale and the memory capacity of the system (the number of memories it can store), with the two quantities becoming linearly related in environments with a high frequency of unreliable memory presentations (see Appendix and *Figure 2—figure supplement 1*).

The quantifications of recall SNR and learnable timescale we present in figures are computed numerically, as deriving exact analytical expressions for learnable timescale is difficult due to the randomness of the interarrival distribution. However, to gain theoretical intuition, we find it useful to consider the following approximation, corresponding to an environment in which memories are reinforced at deterministic intervals of length $\tau$:

$$\text{SNR} \approx \sum_{i=1}^{R} m\left(i\tau\right). \tag{3}$$

This approximation is an upper bound on the true SNR in the limit of small $\epsilon$, and empirically provides a close match to the true dependence of SNR on $R$ (*Figure 2—figure supplement 2*). Using this approximation allows us to provide closed-form analytical estimates of the behavior of SNR and learnable timescale as a function of system and environment parameters.

## Theory of recall-gated consolidation

In the recall-gated consolidation model, the behavior of the STM is identical to that of a model without systems consolidation. The LTM, on the other hand, behaves differently, updating only in response to the subset of memory presentations that exceed a threshold level of recall in the STM. From the perspective of the LTM, this phenomenon has the effect of changing the distribution of interval lengths between repeated reinforcements of a reliable memory. For exponentially distributed interarrival times, the induced effective interarrival distribution in the LTM is also exponential with new time constant $\tau_{\text{LTM}}$ given by

$$\tau_{\text{LTM}} \approx \frac{P\left(I < m^{-1}\left(\theta\right)\right)}{1 - \Phi\left(\theta\right)} \tau. \tag{4}$$

where $I$ is the (stochastic) length of intervals between presentations of the same reliable memory, $\theta$ is the consolidation threshold, and $\Phi$ is the cumulative distribution function of the Gaussian distribution with mean 0 and variance 1. This approximation is valid in the limit of large system sizes $N$, where responses to unreliable memories are nearly Gaussian. For general (non-exponential) interarrival distributions, the shape of the effective LTM interarrival distribution may change, but the above expression for $\tau_{\text{LTM}}$ remains valid.

We note that although the consolidation threshold $\theta$ can be chosen arbitrarily, setting it to too high a value has the effect of reducing the probability with which reliable memories are consolidated, by a factor of $P(I < m^{-1}(\theta))$. For large values of $\theta$ this reduction can become unacceptably small. For a given number of memory repetitions $R$, we restrict ourselves to values of $\theta$ for which the probability that no consolidation takes place after $R$ repetitions is smaller than the allowable probability of error

$\epsilon$. Where $R$ is not explicitly reported, we set $R=2$, which corresponds to analyzing the behavior of the model when a memory is reinforced once following its initial presentation.

## Recall-gated consolidation increases SNR and learnable timescale of memories

For fixed statistics of memory presentations, as the SNR of the STM increases (say, due to increasing $N$), stricter thresholds can be chosen for consolidation which filter out an increasing proportion of unreliable memory presentations, without reducing the consolidation rate of reliable memories (*Figure 4A*, *Figure 4—figure supplement 1*). Consequently, the SNR of the LTM can grow much larger than that of the STM, and the amplification of SNR increases with the SNR of the STM. Notably, the SNR of the LTM in the recall-gated consolidation model also exceeds that of a control model in which STM and LTM modules are both present but do not interact (*Figure 4B*, *Figure 4—figure supplement 1*), which performs comparably to the STM by itself due to the lack of selective consolidation.

We may also view the benefits of consolidation in terms of the learnable timescale of the system. Recall-gated consolidation enables longer learnable timescales, particularly at high target SNRs (*Figure 4C*, *Figure 4—figure supplement 2*). We note that our definition of SNR considers only noise arising from random memory sampling and presentation order. High SNR values may be essential for adequate task performance in the face of additional sources of noise, or when the system is asked to generalize by recalling partially overlapping memory traces (*Benna and Fusi, 2016*).

## Recall-gated consolidation enables better scaling of memory retention with repeated reinforcement

As mentioned previously, higher consolidation thresholds reduce the rate at which reliable memories are consolidated. However, the consolidation rate of *unreliable* memories decreases even more quickly as a function of the threshold (*Figure 4D and E*). Hence, higher thresholds increase the fraction of consolidated memories which are reliable, at the expense of reducing the rate of consolidation into LTM. This tradeoff may be acceptable if reliable memories are reinforced a large number of times, as in this case they can still be consolidated despite infrequent LTM plasticity. In other words, as the number of anticipated repetitions $R$ of a single reliable memory increases, higher thresholds can be used in the gating function, without preventing the eventual consolidation of that memory. Doing so allows more unreliable memory presentations to be filtered out and consequently increases the SNR in the LTM (*Figure 4F*).

Assuming, as we have so far, that reliable memories are reinforced at independently sampled times at a constant rate, we show (calculations in Appendix) that the dependence of learnable timescale on $R$ is linear, regardless of the underlying synaptic model (*Figure 4G*, *Figure 4C*, *Figure 4—figure supplement 3*). Synaptic models with a small number of states, such as binary switch or cascade models, are unable to achieve this scaling without systems consolidation (*Figure 4G*). In particular, the learnable timescale is roughly invariant to $R$ for the binary switch model, and scales approximately logarithmically with $R$ for the cascade model (see Appendix for derivation). Synaptic models employing a large number of internal states (growing exponentially with the intended timescale of memory retention), like the multivariable model of *Benna and Fusi, 2016*, can also achieve linear scaling of learnable timescale on $R$. However, these models still suffer a large constant factor reduction in learnable timescale compared to models employing recall-gated consolidation (*Figure 4G*).

## Consolidation dynamics depend on the statistics of memory recurrence

The benefit of recall-gated consolidation is even more notable when the reinforcement of reliable memories does not occur at independently sampled times, but rather in clusters. Such irregular interarrival times might naturally arise in real-world environments. For instance, exposure to one environmental context might induce a burst of high-frequency reinforcement of the same pattern of synaptic updates, followed by a long drought when the context changes. Intentional bouts of study or practice could also produce such effects. The systems consolidation model can capitalize on such bursts of reinforcement to consolidate memories while they can still be recalled.

To formalize this intuition, we extend our theoretical framework to allow for more general patterns of memory recurrence. In particular, we let $p(I)$ indicate the probability distribution of interarrival intervals $I$ between reliable memory presentations. So far, we have considered the case of reliable

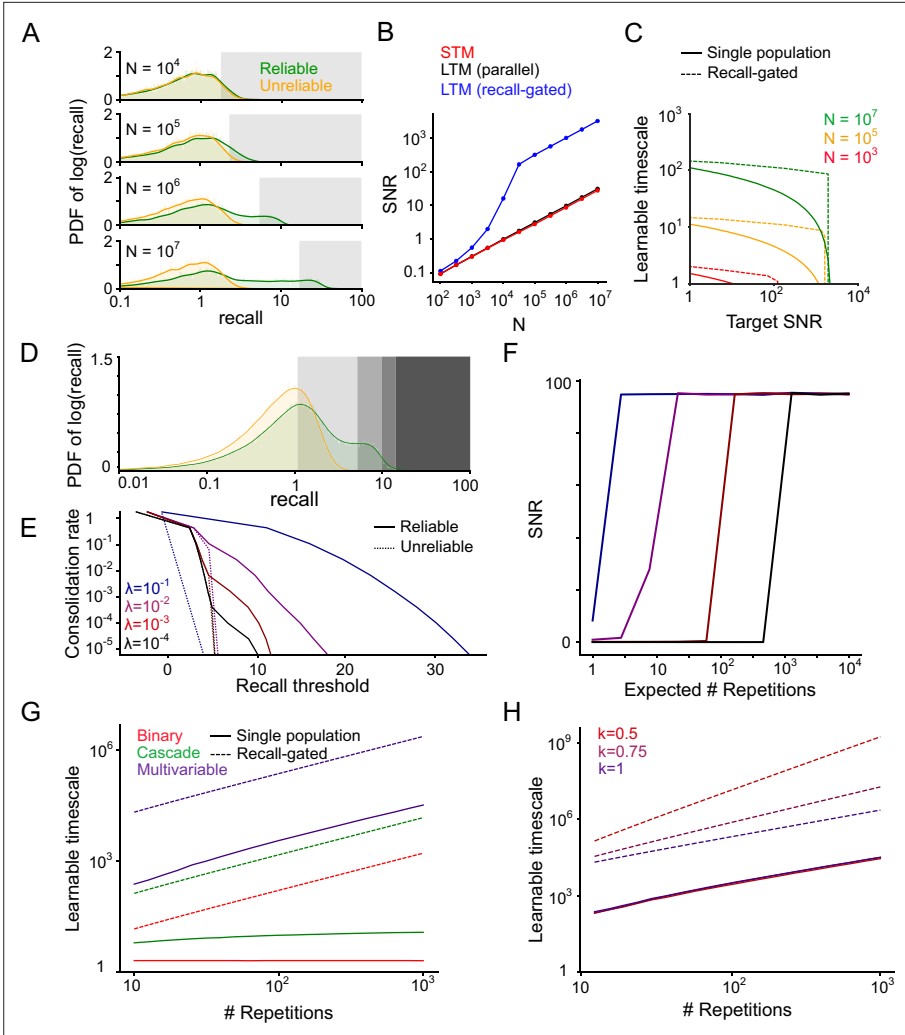

**Figure 4.** Properties of the recall-gated consolidation model. (**A**) Distribution (probability density function) of reliable and unreliable memory overlaps (on a log scale), varying the number of synapses $N$, $\lambda = 10^{-3}$ Shaded regions indicate consolidation thresholds that preserve 10% of reliable memory presentations. Units are standard deviations of the distribution of recall for randomly sampled memories. (**B**) LTM SNR induced by consolidation (with threshold set as in A, to consolidate 10% of reliable memory presentations) as $N$ varies. The parallel model uses aslower learning rate (the value of $p$ in the binary switching synapse model is a factor of 10 smaller) in the LTM than the STM. (**C**) Learnable timescale as a function of target SNR, for several values of N, using the binary switching synapse model with $p = 100/\sqrt{N}$. (**D**) Distribution of reliable and unreliable memory overlaps, with various potential gating thresholds indicated, $N = 10^4$, $\lambda = 10^{-2}$. (**E**) Fraction of memory presentations consolidated (log scale) vs recall threshold for consolidation, $N = 10^4$. (**F**) LTM SNR induced by consolidation vs. the expected number of repetitions before consolidation occurs, $N = 10^4$, same color legend as panel E. Increasing the expected number of repetitions corresponds to setting a more stringent consolidation threshold which filters a higher proportion of reliable memory presentations. (**G**) Learnable timescale at a target SNR of 10 as a function of number of reliable memory repetitions for several underlying synapse models, $N = 10^7$. (**H**) Same as **G**, considering only the multivariable model as the underlying synapse model, and varying the interarrival interval regularity factor $k$.

The online version of this article includes the following figure supplement(s) for figure 4:

**Figure supplement 1.** Effects of varying rate of reliable memory presentations.

**Figure supplement 2.** Comparison of different synaptic learning rules.

**Figure supplement 3.** Same information as *Figure 4G*, varying the population size $N$ and the desired *SNR*.

**Figure supplement 4.** Same information as *Figure 4H*, varying the population size $N$ and the desired *SNR*.

memories whose occurence times follow Poisson statistics, corresponding to an exponentially distributed interval distribution. To consider more general occurrence statistics, we consider a family of interrarival distributions known as Weibull distributions. This class allows control over an additional parameter $k$ which modulates "burstiness" of reinforcement, and contains the exponential distribution as a special case ($k=1$). For $k<1$, reliable memory presentations occur with probability that decays with time since the last presentation. In this regime, the same memory is liable to recur in bursts separated by long gaps (details in Methods).

Without systems consolidation, the most sophisticated synapse model we consider, the multivariable model of *Benna and Fusi, 2016*, achieves a scaling of learnable timescale that is linear with $R$ regardless of the regularity factor $k$. In fact, we show (see Appendix) that the best possible learnable timescale that can achieved by any synaptic consolidation mechanism scales approximately linearly in $R$, up to logarithmic factors. However, for the recall-gated consolidation model, the learnable timescale scales as $R^{1/k}$ when $k \leq 1$ (*Figure 4H*, *Figure 4C*, *Figure 4—figure supplement 3*). In this sense, recall-gated consolidation outperforms any form of synaptic consolidation at learning from irregularly spaced memory reinforcement.

## Alternative gating functions suit different environmental statistics and predict spaced training effects

Thus far, we have considered a threshold gating function, which is well-suited to environments in which unreliable memories are each only encountered once. We may also imagine an environment in which unreliable memories tend to recur multiple times, but over a short timescale (*Figure 5A*, top). In such an environment, the strongest evidence for a memory's reliability is if it overlaps to an *intermediate* degree with the synaptic state (*Figure 5A*, bottom). The appropriate gating function in this case is no longer a threshold, but rather a non-monotonic function of STM memory overlap, meaning that memories are most likely to be consolidated if reinforced at intermediate-length intervals (*Figure 5B*). Such a mechanism is straightforward to implement using neurons tuned to particular ranges of recall strengths. This model behavior is consistent with spaced learning effects reported in flies (*Beck et al., 2000*), rodents (*Glas et al., 2021*), and humans (*Rovee-Collier et al., 1995*; *Verkoeijen et al., 2005*), which all show a characteristic inverted U-shaped dependence of memory performance on spacing interval.

While some synapse-level models (such as the multivariable synapse model of *Benna and Fusi, 2016*) can also give rise to spaced training effects, these effects require that a synapse undergoes few additional potentiation or depression events between the spaced reinforcements (*Figure 5C*, *Figure 5—figure supplement 1*). This is because spacing effects in such models arise when synapse-local variables are saturated, and saturation effects are disrupted when other events are interspersed between repeated presentations of the same memory. Hence, the spacing effects arising from such models are unlikely to be robust over long timescales. Recall-gated systems consolidation, on the other hand, can yield spaced training effects robustly in the presence of many intervening plasticity events.

## Heterogeneous gating functions suit complex environments with multiple memories reinforced at different timescales

Thus far we has assumed a dichotomy between unreliable, one-off memories and reliable memories which recur according to particular statistics. In more realistic scenarios, there will exist many repeatedly reinforced memories, which may be reinforced at distinct timescales. We may be interested in ensuring good recall performance over a distribution of memories with varying recurrence statistics. For concreteness, we consider the specific case of an environment with a large number of distinct reliably reinforced memories, whose characteristic interarrival timescales are log-uniformly distributed. As before, unreliable memories are also presented with a constant probability per timestep.

The recall-gated plasticity model already described, using a threshold function for consolidation, still provides the benefit of filtering unreliable memory traces from the LTM. However, further improved memory recall performance is achieved with a simple extension to the model. The LTM can be subdivided into a set of subpopulations, each with distinct gating functions that specialize for different memory timescales by selecting for different recall strengths (*Figure 5D and E*). That is, one subpopulation consolidates strongly recalled memories, another consolidates weakly recalled

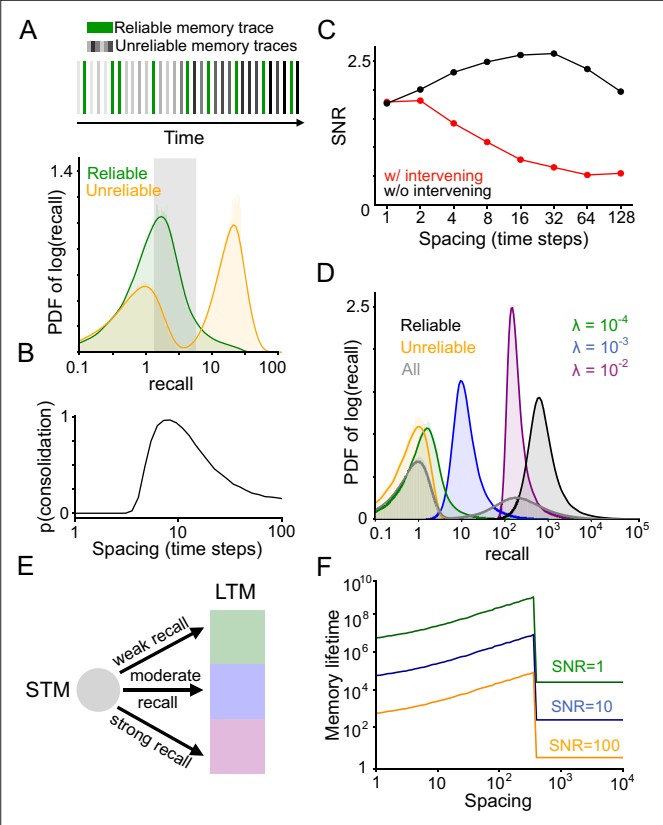

**Figure 5.** Alternative memory gating functions. (**A**) Top: Example sequence of memory presentations where unreliable memories (gray) can repeat multiple times, but only within within a short timescale (note gradient of light to dark). Bottom: Distribution of reliable and unreliable memory overlaps induced by such memory presentation statistics (log scale on x axis). Shaded region indicates overlap values that are at least ten times as likely for reliable memories as for unreliable memories. (**B**) Probability of consolidation, with the gating function chosen such that only overlaps within the shaded region of panel A are consolidated, as a function of interarrival interval. (**C**) SNR at 8 timesteps following 5 spaced repetitions of a memory, with spacing interval indicated on the x axis, for the multivariable synapse model of *Benna and Fusi, 2016* with no systems consolidation. Spaced training effects are present at short timescales, but not if other memories are presented during the interpresentation intervals. (**D**) Distribution of recall strengths corresponding to different kinds of memories, in an environment with many reliable memories. In the environment model, reliable memories are reinforced with different interarrival interval distributions, and the timescales of these distributions for different memories are distributed log-uniformly. The environment also has as a background rate of unreliable memory presentations appearing at fraction 0.9 of timesteps. (**E**) Depiction of a generalization of the model in which memories can be consolidated into different LTM sub-modules, according to gating functions tuned to different recall strengths (intended to target reliable memories with different timescales of recurrence). (**F**) A consequence of the model outlined in panel E is a smooth positive dependence of memory lifetime on the spacing of repetitions, up to some threshold spacing.

The online version of this article includes the following figure supplement(s) for figure 5:

**Figure supplement 1.** Same information as *Figure 5C*, varying the learning rate (scale of potentiation/depression impulses, relative to the maximum/minimum threshold values in the model of *Benna and Fusi, 2016*), and the length of time following spaced training at which the system's recall SNR is evaluated.

**Figure supplement 2.** Same as *Figure 5D* (top row) and *Figure 5F* (bottom row), for different population sizes *N*.

**Figure supplement 3.** Same as *Figure 5D* (top row) and *Figure 5F* (bottom row), for different memory recurrence regularity factors (Weibull distribution parameter *k*).

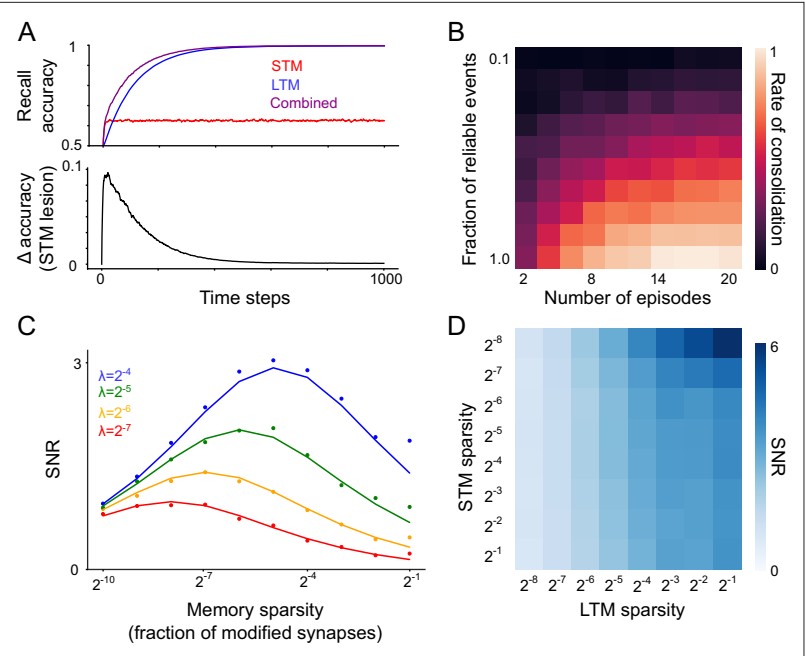

**Figure 6.** Predictions of the model. (**A**) Top: Recall performance for a single reliable memory (true positive rate, with a decision threshold set to yield a 10% false positive rate) as learning progresses. Simulation environment is the same as in *Figure 2C*. $N = 10^3$, $\lambda = 0.25$. Bottom: difference between combined recall performance and LTM-only performance. The STM makes diminishing contributions to recall over time. (**B**) Probability of consolidation into LTM increases with experience and with the reliability of the environment (parameterized here by the recurrence frequency $\lambda$ of reliable memories). Simulation environment is the same as in panel A. (**C**) For a single population of binary synapses (no consolidation) and Poisson memory recurrence, mean SNR as a function of reliable memory recurrence frequency $\lambda$ and memory sparsity $f$. Dots indicate simulation results and solid lines indicate analytical approximation. $N=1,024$. (**D**) For the systems consolidation model using binary synapses, total system SNR ($N=256$) as a function of memory sparsity in the STM and LTM.

memories, and others lie on a spectrum between these extremes. The effect of this arrangement is to assign infrequently reinforced memory traces to subpopulations which experience less plasticity, allowing these traces to persist until their next reinforcement. This heterogeneity of timescales is consistent with observations in a variety of species of intermediate timescale memory traces (*Rosenzweig et al., 1993*; *Cepeda et al., 2008*; *Davis, 2011*).

Studies of spaced training effects have found that the optimal spacing interval during training depends on the interval between training and evaluation (*Cepeda et al., 2006*; *Cepeda et al., 2008*). In particular, the timescale of memory retention is observed to increase smoothly with the spacing interval used during training. Our extended model naturally gives rise to this behavior (*Figure 5F*, *Figure 5—figure supplement 2*, *Figure 5—figure supplement 3*), due to the fact that the lifetime of a consolidated memory scales inversely with the frequency with which memories are consolidated into its corresponding LTM subpopulation.

## Predicted features of memory representations and consolidation dynamics

The recall-gated consolidation model makes a number of key predictions. The most basic consequence of the model is that responsibility for recalling a memory will gradually shift from the STM to the LTM as consolidation progresses, rendering the recall performance of the system increasingly robust to the inactivation of the STM (*Figure 6A*). A more specific prediction of the model is the *rate* of updates to the LTM increases with time, as STM recall grows stronger (*Figure 6B*). The rate of LTM updates also increases with reliability of the environment (operationalized as the proportion of synaptic update events which correspond to reliable memories; *Figure 6B*).

The recall-gated consolidation model also makes predictions regarding neural representations in the STM and LTM. Until now we have assumed that memories consist of balanced potentiation and depression events distributed across the population. However, memories may involve only a sparse subset of synapses, for instance if synaptic plasticity arises from neural activity which is itself sparse. To formalize this notion, we consider memories that potentiate a fraction $f$ of synapses, and a correspondingly modified binary switch plasticity rule such that potentiation activates synapses with probability $p$ and depression inactivates synapses with probability $\frac{f}{1-f}p$. We show analytically (see Appendix) that in the limit of low $f$, the SNR-optimizing choice of $f$ is proportional to the rate $\lambda$ of reliable memory reinforcement (*Figure 6C*). Other factors, such as energetic constraints and noise-robustness, may also affect the optimal coding level. In general, however, our analysis shows that environments with infrequent reinforcement of a given reliable memory incentivize sparser representations. As the effective value of $\lambda$ is amplified in the LTM module, it follows that the LTM benefits from a denser representation than the STM. Interestingly, we also find that the optimal sparsity in the STM decreases when optimizing for the overall SNR of the system—that is, the optimal STM representation is even more sparse in the context of supporting LTM consolidation than it would be in isolation. Taken together, these two effects result in much denser representations being optimal in the LTM than in the STM (*Figure 6D*). One consequence of denser representations is greater generalization in the face of input noise (*Babadi and Sompolinsky, 2014*), implying that an optimal STM/LTM system should employ more robust and generalizable representations in the LTM.

## Discussion

We have presented a theory of systems memory consolidation via recall-gated long-term plasticity, which provides complementary benefits to synaptic consolidation mechanisms in terms of memory lifetime and retrieval accuracy. Its advantage arises from the ability to integrate over the information present in an entire neuronal population, rather than individual synapses, in order to decide which memory traces are consolidated. This capability is important in environments that induce a mixture of reliable and unreliable synaptic updates, in which a system must prioritize which updates to store long-term.

### Experimental evidence for recall-gated consolidation

The recall-gated consolidation model is by design agnostic to the underlying neural circuit and hence potentially applicable to a wide variety of species and brain regions. Here, we summarize evidence consistent with recall-gated consolidation in several model organisms. As our proposal is new, the experiments we describe were not designed to directly test our model predictions, and thus provide incomplete evidence for them. We hope that future work will more directly clarify the relevance of our model to these systems as well as others, the mechanisms by which it is implemented, and the shortcomings it may have in accounting for experimental results.

#### Associative learning in insects

In the *Drosophila* mushroom body, plasticity is driven by activity of dopamine neurons innervating a collection of anatomically defined compartments. These contain mushroom body output neurons (MBONs) that drive learned behavioral responses, such as approach or avoidance, to sensory stimuli (*Aso et al., 2014*). The compartments are grouped into anatomically defined lobes referred to by Greek letters: $\gamma, \alpha, \beta, \alpha', \beta'$. In general the $\gamma$ lobe compartments are implicated in STM while the $\alpha/\beta$ compartments are implicated in LTM (*Aso et al., 2014*). Mushroom body dopamine neurons receive a wide variety of inputs, including from MBONs themselves (*Li et al., 2020*). Such inputs provide a substrate by which long-term learning can be modulated by the outputs of short-term pathways. To implement recall-gated consolidation, the activity of dopamine neurons modulating plasticity in LTM compartments should be gated by learning in corresponding short-term pathways. A recent study found an instance of this motif (*Awata et al., 2019*). Short-term aversive learning decreases the activity of the $\gamma 1$ MBON (implicated in short-term aversive memory). This $\gamma 1$ MBON is inhibitory and synapses onto a dopamine neuron innervating the $\alpha 2$ compartment (which is associated with long-term aversive learning). Thus, short-term aversive learing in the $\gamma 1$ MBON disinhibits the $\alpha 2$ dopamine neuron, allowing for learning to proceed in the LTM $\alpha 2$ compartment. This circuit is a

precise mechanistic implementation of our recall-gated consolidation model. More work is needed to determine if other examples of this motif can be found in *Drosophila* or other insects.

### Motor learning

Several lines of work have indicated that the neural substrate of motor skills can shift with practice. In songbirds, learned changes to song performance are initially driven by a cortico-basal ganglia circuit called the anterior forebrain pathway (AFB) but eventually are consolidated into the song motor pathway (SMP) and become AFB-independent (*Andalman and Fee, 2009*; *Warren et al., 2011*). Using transient inactivations of LMAN, a region forming part of the AFB, a recent study quantified the degree of AFB-to-SMP consolidation over time and found that it strongly correlated with the bird's motor performance at the time (*Tachibana et al., 2022*). Although this finding does not establish the mechanism for this phenomena, the behavioral result is consistent with our model's prediction that the *rate* of consolidation should increase as learning progresses in the short-term pathway.

A related motor consolidation process has been observed during motor learning in rats. Experiments have shown that motor cortex disengages from heavily practiced skills (*Kawai et al., 2015*; *Hwang et al., 2019*), transferring control at least in part to the basal ganglia (*Dhawale et al., 2019*; *Dhawale et al., 2021*), and that the degree of cortical disengagement tracks motor performance, as measured by the variability of learned trajectories (*Hwang et al., 2021*). This finding is broadly consistent with recall-consolidation, with short-term learning being mediated by motor cortex and long-term learning being mediated by basal ganglia. However, we note that unlike in the song learning study referenced above, it neither confirms nor rejects our stronger prediction that the *rate* (rather than overall extent) of motor consolidation increases with learning.

### Spatial learning and hippocampal replay

Hippocampal replay is thought to be crucial to the consolidation of episodic memories to cortex (*Carr et al., 2011*; *Ólafsdóttir et al., 2018*). Replay has many proposed computational functions, such as enabling continual learning (*van de Ven et al., 2020*), or optimizing generalization performance (*Sun et al., 2021*), which are outside the scope of our model. However, under the assumption that replay enables LTM storage in cortex, the recall-gated consolidation model makes predictions about which memories should be replayed—namely, replay should disproportionately emphasize memories that are familiar to the hippocampus. That is, we would predict more frequent replay of events or associations that are frequently encountered than of those that were experienced only once, or unreliably.

Recent experimental work supports this hypothesis. A recent study found that CA3 axonal projections to CA1, those that respond visual cues associated with a fixed spatial location are recruited more readily in sharp-wave ripple events than those that respond to the randomly presented cues (*Terada et al., 2022*). This observation is consistent with our model's prediction that repeatedly experienced patterns of activity are more likely to be consolidated, though other interpretations are possible. Earlier work found that sharp-wave ripple events occur more frequently during maze navigation sessions with regular trajectories, and increase in frequency over the course of session, similar to the behavior of our model in *Figure 6B*; *Jackson et al., 2006*. Thus, existing evidence suggests that hippocampal replay is biased toward familiar patterns of activity, consistent with a form recall-gated consolidation. Other experiments provide preliminary evidence for signatures of such a bias in cortical plasticity. For instance, fMRI study of activity in hippocampus and posterior parietal cortex (PPC) during a human virtual navigation experiment found that that the recruitment of PPC during the task, which was linked with memory performance, tended to strengthen with experience in a static environment, but did not strengthen when subjects were exposed to an constantly changing environment, consistent with consolidation of only reliable memories (*Brodt et al., 2016*).

### Comparison with synaptic consolidation mechanisms

Recall-gated consolidation improves memory performance regardless of the underlying synapse model (*Figure 4*), indicating that its benefits are complementary to those of synaptic consolidation. Our theory quantifies these benefits in terms of the scaling behavior of the model's maximum learnable timescale with respect to other parameters. First, for any underlying synapse model, recall-gated consolidation allows the learnable timescale to decay much more slowly as a function of the desired SNR of memory storage. Second, recall-gated consolidation achieves (at worst) linear scaling of

learnable timescale as a function of the number of memory reinforcements $R$. For models with a fixed, finite number of internal states per synapse, this scaling is at best logarithmic. Our results therefore illustrate that systems-level consolidation mechanisms allow relatively simple synaptic machinery to support LTM storage. We note that more sophisticated synaptic models, which involve a large number of internal states that scales with the memory timescale of interest (*Benna and Fusi, 2016*), can also achieve linear scaling of learnable timescale with $R$ (although recall-gated consolidation still improves their performance by a large constant factor). However, for environmental statistics characterized by concentrated bursts of repeated events separated by long gaps, recall-gated consolidation achieves superlinear power-law scaling, which we showed is not achievable by any synapse-local consolidation mechanism.

Our model provides an explanation for spaced training effects (*Figure 5*) based on optimal gating of LTM consolidation depending on the recurrence statistics of reliable stimuli. It is important to note that, depending on the specific form of internal dynamics present in individual synapses, synaptic consolidation models can also reproduce spacing effects. For example, the initial improvement of memory strength with increased spacing arises in the model of *Benna and Fusi, 2016* due to saturation of fast synaptic variables, meaning that the timescale of these internal variables determines optimal spacing, and that intervening stimuli can block the effect by preventing saturation (*Figure 5*). In contrast, in our model this timescale is set by population-level forgetting curves, rendering spacing effects robust over long timescales and in the presence of intervening events. It is likely that mechanisms at both the synaptic and systems level contribute to spacing effects; our results suggest that effects observed at longer timescales are likely to arise from memory recall mechanisms at the systems level.

## Other models of systems consolidation

Unlike previous theories, our study emphasizes the role of repeated memory reinforcement and selective consolidation. As such, our model has novel capabilities, but also has limitations compared to other models of consolidation. As the key insight of our model is distinct from most other theoretical work on the subject, we believe that future work will be able to fruitfully integrate the notion of recall-gated plasticity into other models of consolidation and attain the benefits of both.

Much prior work focuses on consolidation via hippocampal replay. Prior work has proposed that replay (or similar mechanisms) can prolong memory lifetimes (*Shaham et al., 2021*; *Remme et al., 2021*), alleviate the problem of catastrophic forgetting of previously learned information (*van de Ven et al., 2020*; *González et al., 2020*; *Shaham et al., 2021*), and facilitate generalization of learned information (*McClelland et al., 1995*; *Sun et al., 2021*). One prior theoretical study (*Roxin and Fusi, 2013*), which uses replayed activity to consolidate synaptic changes from short to long-term modules, explored how systems consolidation extends forgetting curves. Unlike our work, this model (and related work, such as that of *Brea et al., 2023*) essentially models consolidation as 'copying' synaptic weights from one system to another. While such a mechanism has potentially useful consequences, such as enabling decoding of the age of a memory (*Brea et al., 2023*), it does not involve gating of memory consolidation, and consequently provides no additional benefit in consolidating repeatedly reinforced memories. Our model is thus distinct from, but also complementary to, these prior studies. In particular, recall-gated consolidation can be implemented in real-time, without replay of old memories. However, as discussed above, selective replay of familiar memories is one possible implementation of recall-gated consolidation. Selective replay is a feature of some of the work cited above (*Shaham et al., 2021*; *Sun et al., 2021*), which suggests it can provide advantages for retention and generalization (*Shaham et al., 2021*; *Sun et al., 2021*).

Other work has proposed models of consolidation, particularly in the context of motor learning, in which one module 'tutors' another to perform learned behaviors by providing it with target outputs (*Murray and Escola, 2017*; *Teşileanu et al., 2017*). *Murray and Escola, 2020* proposes a fast-learning pathway (which learns using reward or supervision) which tutors the slow-learning long-term module via a Hebbian learning rule. In machine learning, a similar concept has become popular (typically referred to 'knowledge distillation'), in which the outputs of a trained neural network are used to supervise the learning of a second neural network on the same task (*Hinton et al., 2015*; *Gou et al., 2021*). Empirically, this procedure is found to improve generalization performance and enable the use of smaller networks. Our model can be interpreted as a form of partial tutoring of

the LTM by the STM, as learning in the LTM is partially dictated by outputs of the STM. In this sense, our work provides a theoretical justification for the use of tutoring signals between two neural populations.

## Limitations and future work

In addition to motivating new experiments to test the predictions of a recall-gated consolidation model, our work leaves open a number of theoretical questions that future modeling could address. Our theory assumes fixed and random representations of memory traces. Subject to this assumption, we showed that STM benefits from sparser representations than LTM. In realistic scenarios, synaptic updates are likely to be highly structured, and the optimal representations in each module could differ in more sophisticated ways. Moreover, adapting representations online—for instance, in order to decorrelate consolidated memory traces—may improve learning performance further. Addressing these questions requires extending our theory to handle memory statistics with nontrivial correlations. Another possibility we left unaddressed is that of more complex interactions between memory modules—for instance, reciprocal rather than unidirectional interactions—or the use of more than two interacting systems.

Finally, in this work we considered only a limited family of ways in which long-term consolidation may be modulated—namely, according to threshold-like functions of recall in the short-term pathway. Considering richer relationships between recall and consolidation rate may enable improved memory performance and/or better fits to experimental data. Moreover, in real neural circuits, additional factors besides recall, such as reward or salience, are likely to influence consolidation as well. For instance, a sufficiently salient event should be stored in LTM even if encountered only once. Furthermore, while in our model familiarity drives consolidation, certain forms of novelty may also incentivize consolidation, raising the prospect of a non-monotonic relationship between consolidation probability and familiarity. Unlike our notion of recall, which can be modeled in task-agnostic fashion, the impact of such additional factors on learning likely depends strongly depend on the behavior in question. Our work provides a theoretical framework that will facilitate more detailed models of the rich dynamics of consolidation in specific neural systems of interest.

## Methods
### Theoretical framework

We consider a population of $N$ synapses, indexed by $i \in \{1, 2, ..., N\}$ each with a synaptic weight $w_i \in \mathbb{R}$. The set of synaptic weights across the population can be denoted by the vector $\mathbf{w} \in \mathbb{R}^N$. The synapses may retain additional information besides strength as well; if each synapse carries $d$-dimensional state information in addition to its strength, the synaptic state can be written as $\mathbf{w_i} \in \mathbb{R}^d$, with the scalar synaptic strengths $w_i \in \mathbb{R}$ defined as a function of the high-dimensional state $\mathbf{w_i}$. We define memories as patterns of target synaptic weights, following prior work (***Benna and Fusi, 2016***; ***Fusi et al., 2005***). More specifically, we model each memory as a vector $\mathbf{w}^* \in \mathbb{R}^N$. By defining memories in this fashion, our analysis can remain agnostic to the network architecture and plasticity rule that give rise to synaptic modifications. We will typically model memories as binary potentiation/depression events for simplicity, but in principle, memories can be continuous valued. Synaptic are updated by memories according to a plasticity rule $(\tilde{\mathbf{w}}_i)_{new} = \ell(\tilde{\mathbf{w}}, m_i)$, which maps the synaptic state at the time of a memory event to the subsequent synaptic state.

For theoretical calculations, we assume as in prior work (***Fusi et al., 2005***; ***Benna and Fusi, 2016***), that the components of each memory $\mathbf{w}^*$ are independent and uncorrelated with those of other memories (although this assumption is violated in our task learning simulations). We also assume for simplicity that memories are mean-centered so that $E[\mathbf{w} \cdot \mathbf{w}^*_{\text{rand}}] = 0$ over randomly sampled memories $\mathbf{w}^*_{\text{rand}}$.

We define the recall strength associated with memory as the overlap $r = \mathbf{w} \cdot \mathbf{w}^*$. This definition reflects an 'ideal observer' perspective, as it requires direct and complete access to the state of the synaptic population. The ideal observer perspective provides an upper bound on the recall performance of a real system, and should be a fairly good approximation assuming that memory readout mechanisms are sophisticated enough. We are particularly interested in the normalized recall strength

$$\text{SNR} = \frac{\mathbf{w} \cdot \mathbf{w}^*}{\sqrt{E_{\mathbf{w}_{\text{rand}}^*} \left[ \left( \mathbf{w} \cdot \mathbf{w}_{\text{rand}}^* \right)^2 \right]}},$$ 
(5)

where the expectation is taken over randomly sampled memories $\mathbf{w}_{\text{rand}}^*$. We refer to this quantity as the signal-to-noise ratio (SNR) of memory recall.

## Synaptic models and plasticity rules

In this paper, we primarily consider three synaptic models and corresponding plasticity rules, taken from prior work.

The first and simplest is is a 'binary switch' model in which synapses take on binary ( ± 1) values and stochastically activate (resp. inactivate) in response to positive (resp. negative) values of a memory vector with probability $p$ (**Amit and Fusi, 1994**). No auxiliary state variables are used in this model.

The second is the 'cascade' model of **Fusi et al., 2005**, in which synapses are modeled as a Markov chain with a finite number 2 k of discrete states with transition probabilities dependent on the kind of memory event (potentiation or depression). Half the states (states $a_1, ..., a_k$) are considered potentiated (strength +1) and half (states $b_1, ..., b_k$) are depressed (strength –1). Intuitively, states of the same potentiation level correspond to different propensities for plasticity in the synapse, enabling a form of synaptic consolidation. Formally, for $i<k$, the potentiated state $a_i$ (resp. depressed state $b_i$) transitions to state $a_{i+1}$ (resp. $b_{i+1}$) with probability $\frac{\alpha^i}{1-\alpha}$ following a potentiation (resp. depression) event. And for $i<k$, the potentiated state $a_i$ (resp. depressed state $b_i$) transitions to state $b_1$ (resp. $a_1$) with probability $\alpha^{i-1}$ following a depression (resp. potentiation) event. For $i=k$ this latter transition occurs with probability $\frac{\alpha^{i-1}}{1-\alpha}$; as described in **Fusi et al., 2005**, this choice is made for convenience to ensure equal occupancy of the different synaptic states. We assume $\alpha = 0.5$ throughout.

The third synaptic model is the model of **Benna and Fusi, 2016**, which we refer to as the 'multi-variable' model. In this model, synapses are described by a chain of $m$ interacting continuous-valued variables $u_1, ..., u_m$, the first of which corresponds to synaptic strength. Potentiation and depression events increment or decrement the value of the first synaptic variable, and a set of difference equations governs the evolution of the multidimensional state at each time step:

$$\frac{du_i}{dt} = n^{-2i+2} \alpha \left( u_{i-1} - u_i \right) - n^{-2i+1} \alpha \left( u_i - u_{i+1} \right),$$
(6)

where $n$ and $\alpha$ a parameter of the model (we assume $n = 2, \alpha = 0.5$ throughout). This model also provides the ability for synapses to store information at different timescales, due to the information retained in auxiliary variables.

## Model implementation for example tasks

### Supervised Hebbian learning

We simulated a single-layer feedforward network with a population of N=1,000 input neurons (activity denoted by $\mathbf{x}$) and a single output neuron, (activity denoted by $y$), connected with a 1×N binary weight matrix $\mathbf{W}$, such that $\hat{\mathbf{y}} = \mathbf{Wx}$. In each simulation, a set of $P$=20 reliable stimuli were randomly generated, which corresponded to binary ($\pm\frac{1}{N}$) random $N$-dimensional activity patterns in the input neurons. Note that due to the scaling of the inputs and use of binary synaptic weights, the activity $y$ is constrained to lie in the interval $[-1, 1]$. Each reliable stimulus was associated with a randomly chosen (but consistent across the simulation) label $y$, 1 or –1. At each time step, one of the reliable stimuli (along with its label) was presented to the network with probability $\lambda_i = 0.01$ for all $i = 1, ..., P$. Otherwise (with probability $1 - \sum_i \lambda_i$), a randomly sampled unreliable stimulus was presented with a randomly chosen label. Memory vectors (written as matrices since the synaptic weights are interpreted as a matrix) were given by a Hebbian learning rule $\mathbf{W}^* = y\mathbf{x}^T$, corresponding to the product of the binary input neuron activity and the corresponding label. Learning followed the binary switch rule with $p$=0.1; that is, positive entries in the memory vector resulted in potentiation with probability $p$, and likewise for negative entries and depression. At each timestep, the product of the STM output and the ±1 label was computed, and if it exceeded the consolidation threshold $\theta = 0.125$, plasticity was permitted in the LTM network.

## Reinforcement learning

We used the same setup as in the supervised learning task, with the following modifications. Instead of a single readout, the network had $A=3$ output neurons corresponding to different possible actions. The activity of the output neuron $a$ (denoted by $\pi_a$) represented the unnormalized log probabiliy of taking action $a$: $p(a) = \frac{e^{\beta \pi_a}}{\sum_j e^{\beta \pi_j}}$, where $\beta$ is a parameter controlling the stochasticity of the action selection (we set $\beta=10$ in our simulations). For the purposes of learning, the STM outputs were used to compute action probabilities, but both the STM and the LTM were evaluated throughout training. Each of the 5 reliable stimuli was associated with a correct action. Taking the correct action yielded a reward of 1, while taking the other action yielded a reward of 0. Unreliable stimuli were associated with randomly sampled correct actions. Memory vectors were derived from the following three-factor learning rule: $\mathbf{W}^* = \text{reward} \cdot (\mathbf{a}\mathbf{x}^T)$, where $\mathbf{a}$ is a one-hot vector with a value of 1 at the index of the chosen action. At each timestep the product $\pi_a \cdot \text{reward}$ was computed, and if it exceeded the consolidation threshold, plasticity was permitted in the LTM network. All other parameters were the same as in the supervised learning simulation.

## Unsupervised Hebbian learning

We simulated two recurrent neural networks with $N=1,000$ binary neurons each and with binary recurrent weight matrices $\mathbf{W}_{\text{STM}}$ and $\mathbf{W}_{\text{LTM}}$, respectively. Memories consisted of binary (entries equal to $\pm\frac{1}{N}$) random $N$-dimensional vectors that provided direct input $\mathbf{x}$ to the network neurons at each timestep. The network state $\mathbf{h}$ evolved for $T=5$ timesteps according to the following dynamics equation:

$$\mathbf{h}_{t+1} = \mathbf{W}\phi\left(\mathbf{h}\right) + \mathbf{x}, \tag{7}$$

where $\phi$ is a binary threshold nonlinearity with threshold set so that 50% of neurons were active at each time step (corresponding to a mechanism that normalizes activity across the network). The weights $\mathbf{W}$ of the network were binary and initialized as binary random variables with equal on/off probability. On each trial a stimulus was presented, which with probability $\lambda = 0.25$ was a (randomly sampled but consistent) reliable stimulus, and otherwise was a newly randomly sampled unreliable stimulus. The network weights $W_{ij}$ were subjected to potentiation events when $x_i$ and $x_j$ were both active at $t=0$, and otherwise subjected to depression events. Synaptic updates followed the binary switch rule with probability $P=1.0$.

Additionally, a set of $N$ weights $\mathbf{u}$ connected the STM neurons to a single readout neuron that measured familiarity. These weights were also binary and updated according to their own memory vector $\mathbf{u}^*$. They experienced candidate potentiation/depression events when their corresponding stimulus input neuron was active/inactive, respectively (i.e. the memory entry $u_i^*$ was equal to $x_i$). These weights followed the binary switch rule with probability $p=1.0$.

Plasticity in the LTM proceeded according to the same rule as in the STM but was gated by recall strength $r = \mathbf{u} \cdot \mathbf{x}_{\text{STM}}$, according to a threshold function with threshold equal to 0.25.

The performance of the network was determined by presenting noise-corrupted versions of the single reliable stimulus and measuring the correlation between the network state and the uncorrupted memory after $T=5$ time steps. The corrupted patterns were obtained by adding Gaussian noise of variance $\frac{1}{N} = 0.001$ to the ground-truth pattern, and binarizing the result by choosing the fraction 0.5 of neurons with the highest values to be active.

## Forgetting curves for different synaptic plasticity rules

Prior work (*Fusi et al., 2005*; *Benna and Fusi, 2016*) has considered environments in which a given memory is presented to the system only once. In this case, the performance of a single population of synapses with a given plasticity rule depends crucially on the memory trace function $m(t)$. This is defined as

$$m\left(t\right) = \frac{\mathbf{w}\left(t\right) \cdot \mathbf{w}^*}{\sqrt{E_{\mathbf{w}_{\text{rand}}^*}\left[\left(\mathbf{w}\left(t\right) \cdot \mathbf{w}_{\text{rand}}^*\right)^2\right]}}, \tag{8}$$

the recall SNR at time $t$ for a memory $\mathbf{w}^*$ presented at $t=0$, assuming randomly sampled memories have been presented in the intervening timesteps. For the binary switch model, $m(t) \approx \sqrt{N}pe^{-pt}$. More sophisticated synaptic models, like the cascade and multivariable models, can achieve power-law scalings (**Fusi et al., 2005**; **Benna and Fusi, 2016**). The key feature of these models that enables power-law forgetting is that their synapses maintain additional information besides their weight, which encodes their propensity to change state. In this fashion, memories can be consolidated at the synaptic level into more stable, slowly decaying traces. The cascade model of Fusi et al. achieves

$$m\left(t\right) \approx \frac{\sqrt{N}}{t \log T} e^{-t/T} \tag{9}$$

for some characteristic timescale $T$ which can be chosen as a model parameter. Hence, its performance is upper bounded by

$$m\left(t\right) \approx \frac{\sqrt{N}}{t \log t}. \tag{10}$$

The model of Benna and Fusi can achieve

$$m\left(t\right) \approx \sqrt{\frac{N}{t \log T}} e^{-t/T} \tag{11}$$

which is upper bounded by

$$\sqrt{\frac{N}{t \log t}}. \tag{12}$$

Benna and Fusi also show that $\sqrt{N/t}$ scaling is an upper bound on the performance of any synapse model with finite dynamic range.

## Implementation of SNR and learnable timescale computations

To compute recall strengths associated with single synaptic populations, we first sampled interarrival intervals $I$ from the environmental statistics $p(I)$. Given a number of repetitions $R$, we computed recall strength samples $r' = \sum_i^R m(t_i)$, where $m$ is the forgetting curve associated with the underlying synaptic plasticity rule, $t_i = \sum_{j \geq i} I_j$, and the $I_j$ are independent samples from $p(I)$. We scaled recall strengths by a factor of $\frac{1}{\sqrt{N}}$ to compute the recall SNR (an approximation that is exact in the large-$N$ limit).

To compute recall strengths associated with the recall-gated consolidation model, we repeated the above procedure using a new interarrival distribution $p(I_{\text{LTM}})$ induced by the gating model. The induced distribution $I_{\text{LTM}}$ is obtained by drawing as samples the lengths of intervals between consecutive interarrival interval samples for which the corresponding recall SNR in the STM exceeds the gating threshold $\theta$ (corresponding to the interval between consolidated reliable memory presentations), and rescaling it by the fraction of unreliable memories that are consolidated. Strictly speaking, in the general case this distribution is nonstationary, as the probability of STM recall exceeding the threshold can change as synaptic updates accumulate across repetitions for sophisticated synapse models like that of **Benna and Fusi, 2016**. We adopt a conservative approximation that ignores such effects and thus slightly underestimates the rate of consolidation when such synaptic models are used (and consequently underestimates the SNR and learnable timescale of the recall-gated consolidation model). With this approximation, the random variable $I_{\text{LTM}}$ is defined as as the following mixture distribution

$$\tilde{I}_{\text{LTM}} = \sum_{i=1}^{j} I_i \quad \text{w.p. } q\left(1-q\right)^{j-1},$$
$$I_{\text{LTM}} = \frac{\tilde{I}_{\text{LTM}}}{\sum_{t=1}^{\tilde{I}_{\text{LTM}}} 1\left[\zeta_t > \theta\right]}, \tag{13}$$

where each $I_i \sim p(I)$, $\zeta_t \sim N(0, 1)$, $q$ indicates the probability of a reliable memory presentation inducing consolidation, and $1[\cdot]$ denotes an indicator function, equal to 1 or 0 depending on whether

the condition is met. The value of $j$ corresponds to the number of reinforcements that go by between instances of consolidation. For sufficiently large $\tau$ this distribution can be approximated by

$$I_{\text{LTM}} = \frac{\sum_{i=1}^{j} I_i}{1 - \Phi(\theta)} \quad \text{w.p. } q(1-q)^{j-1}, \tag{14}$$

where $\Phi$ is the CDF of the standard normal distribution. For large $N$, the probability of consolidation $q = P(I < m^{-1}(\theta))$.

We note that for an exponential interarrival distribution with mean $\tau$, the induced distribution of $I_{\text{LTM}}$ is also exponential, with mean $\tau_{\text{LTM}} = \frac{P(I<m^{-1}(\theta))}{1-\Phi(\theta)}$. This is because the sums of $j$ independent samples $I_i$ are distributed according to a Gamma distribution with shape parameter $j$, and the mixture of such Gamma distributions with geometrically distributed mixture weights $p(j) = q(1-q)^{j-1}$ is itself an exponential distribution with mean $\tau/q$.

For a given number of expected memory repetitions $R$, the gating threshold $\theta$ was set such that at least one of the $R$ repetitions would be consolidated with probability $1 - \epsilon$, $\epsilon = 0.1$. Where $R$ is not reported, we assume it equal to 2, the minimum number of repetitions for the notion of consolidation to be meaningful in our model.

To compute learnable timescales, we repeated the above SNR computations over a range of mean interarrival times $\tau = E[I]$, keeping the interarrival distribution family (Weibull distributions with a fixed value of $k$, see below) constant. We report the maximum value of $\tau$ for which the SNR exceeds the designated target threshold with probability $1 - \epsilon$, $\epsilon = 0.1$.

Throughout, for our interarrival distributions we use Weibull distributions with regularity parameter $k$. The corresponding cumulative distribution function is

$$P(I \leq t) = 1 - e^{-\left(t\Gamma(1+1/k)/\tau\right)^k}. \tag{15}$$

where $\tau = E[I]$, and $\Gamma$ is the Gamma function. In the case $k=1$, this reduces to an exponential distribution of interarrival intervals, which corresponds to memory reinforcements that occur according to a Poisson process with rate $\lambda = 1/\tau$. In the limit $k \to \infty$, it corresponds to interarrival intervals of deterministic length $\tau$. For $k<1$, the interarrival distribution is 'bursty', with periods of dense reinforcement separated by long gaps.

## Spacing effect simulations

We simulated the multivariable synapse model of *Benna and Fusi, 2016*, in which each synapse is described by $m$ continuous-valued dynamical variables $u_1, ..., u_m$ which evolve as follows:

$$\frac{du_i}{dt} = n^{-2i+2}\alpha(u_{i-1} - u_i) - n^{-2i+1}\alpha(u_i - u_{i+1}). \tag{16}$$

For the first variable $u_1$, in place of $u_{i-1}$ we substitute components $m_j$ of the memory traces. For the last variable $u_m$, in place of $u_{i+1}$ we substitute 0. The strength of each synapse corresponds to the value of its first dynamical variable. For our simulations, we chose $m=10$ dynamical variables, n=2, $\alpha = 0.5$, and N=400 synapses. The value of $\alpha$ is also varied in *Figure 5—figure supplement 3*. A spacing interval $\Delta$ was selected and a randomly drawn reliable memory was presented at $\Delta$-length intervals (the same pattern at each presentation). In the case without intervening memories, the dynamics of each synapse ran unimpeded between these presentations. In the case with intervening memories, new randomly drawn patterns were presented to the system at each timestep between the reliable memory presentations. Each pattern was drawn with values equal to ± 1/2, with equal probability.

## Generalized model with multiple memory timescales

In our generalized environment model, the environment contains a variety of distinct reliable memories $x_i$ which recur with Poisson statistics at a variety of rates $\lambda_i$. Timescales $\tau_i = 1/\lambda_i$ are distributed as $p(\log \tau) \sim [0, A]$ where $A$ is a large constant. This corresponds to the value of $\log \lambda$ being uniformly distributed in $[-A, 0]$, or equivalently to $p(\lambda) \sim 1/\lambda$ and bounded between $e^{-A}$ and 1. The environment also contains an additional fraction of unreliable memories as before, sampled randomly and presented with a fixed probability at each timestep. The natural generalization of learnable timescale

to this setting is the maximum interarrival interval timescale for which the lifetime of a corresponding memory (the time following last reinforcement its recall strength decays to an SNR below the target SNR) exceeds that timescale.

The distribution of interarrival intervals for memory $i$ is

$$p\left(I_i\right) = \lambda_i e^{-\lambda_i I_i}. \tag{17}$$

Integrating across the distribution of $\lambda$, we get the distribution of interarrival intervals for reliable memories observed by the system:

$$
\begin{aligned}
p_{\text{reliable}}\left(I\right) &\approx E_{\log\lambda\sim U\left(\left[-A,0\right]\right)}\left[\lambda e^{-\lambda I}\right] \\
&\approx \int_{\lambda=e^{-A}}^{\lambda=1} \frac{1}{\lambda}\lambda e^{-\lambda I} d\lambda \\
&\approx \int_0^1 e^{-\lambda I} d\lambda \\
&\approx \int_0^1 e^{-\lambda I} d\lambda \\
&= \frac{1-e^{-I}}{I} \\
&\approx 1/I
\end{aligned} \tag{18}
$$

for large $I$.

The full distribution of interval strengths (including unreliable memories) is a mixture of $p_{\text{reliable}}$ and a delta function at $I = \infty$, with the latter's weight corresponding to the probability with which an unreliable memory is sampled at a given timestep (in our simulations we chose 0.9).

From here we can compute a distribution of STM recall strengths $r$

$$p\left(r\right) = \int p\left(I\right) m\left(I\right) dI. \tag{19}$$

We simulated a model in which an ensemble of LTM subpopulations are assigned gating functions $g_i(r)$ equal 1 for $\log r \in [A_i, A_{i+1}]$ and 0 elsewhere, with the $A_i$ spaced evenly over $\left[0, \frac{1}{2}\log N\right]$. The expected lifetime of a memory reinforced with a given interval $I'$ is given approximately by the STM lifetime divided by the fraction of memory presentations for which the recall strength lies in the same interval as $m(I')$. This quantity reflects the proportion of memories presentations that are consolidated into the same LTM subpopulation as the memory in question.

## Acknowledgements

We thank Stefano Fusi and Samuel Muscinelli for helpful discussions and comments on the manuscript. ALK and JL were supported by the Gatsby Charitable Foundation, NSF award DBI-1707398. JL was also supported by the DOE CSGF (DE-SC0020347). ALK was also supported by the Burroughs Wellcome Foundation, the McKnight Endowment Fund, the Mathers Foundation, and NIH award R01EB029858.

## Additional information

### Funding

| Funder | Grant reference number | Author |
|---|---|---|
| Gatsby Charitable Foundation | | Jack W Lindsey<br>Ashok Litwin-Kumar |
| National Science Foundation | DBI-1707398 | Jack W Lindsey<br>Ashok Litwin-Kumar |
| Department of Energy | CSGF (DE-SC0020347) | Jack W Lindsey |
| Burroughs Wellcome Fund | | Ashok Litwin-Kumar |
| McKnight Foundation | | Ashok Litwin-Kumar |

| Funder | Grant reference number | Author |
|---|---|---|
| Mathers Foundation | | Ashok Litwin-Kumar |
| National Institutes of Health | R01EB029858 | Ashok Litwin-Kumar |

The funders had no role in study design, data collection and interpretation, or the decision to submit the work for publication.

## Author contributions

Jack W Lindsey, Conceptualization, Formal analysis, Writing - original draft, Writing - review and editing; Ashok Litwin-Kumar, Conceptualization, Writing - review and editing

## Author ORCIDs

Jack W Lindsey http://orcid.org/0000-0003-0930-7327
Ashok Litwin-Kumar http://orcid.org/0000-0003-2422-6576

Reviewer #2 (Public Review): https://doi.org/10.7554/eLife.90793.3.sa1
Reviewer #3 (Public Review): https://doi.org/10.7554/eLife.90793.3.sa2
Author response https://doi.org/10.7554/eLife.90793.3.sa3

---

# Additional files

## Supplementary files
• MDAR checklist

## Data availability
The current manuscript is a theoretical study, so no data have been generated for this manuscript.

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

# Appendix 1

## Derivation of recall strength quantity for specific plasticity rules

In the following derivations, we derive the recall factors for various learning rules, which correspond to different choices of $\mathbf{W}^*$. The recall factor is defined as the elementwise dot product between $\mathbf{W}$ and $\mathbf{W}^*$, which we denote as $\mathbf{W} \cdot \mathbf{W}^*$. We make use of the fact that this elementwise dot product between the matrices is equal to $\text{Tr}[\mathbf{W}^T \mathbf{W}^*]$.

### Supervised Hebbian learning

Let $\mathbf{x}$ be the input population activity, $\mathbf{W}$ be the prediction weights, $\hat{\mathbf{y}} = \mathbf{W}\mathbf{x}$ the output population activity (predicted probabilities), and $\mathbf{y}$ indicate ground-truth target values. A supervised Hebbian plasticity rule gives rise to a memory vector (interpreted here as a matrix since the synaptic weights form a matrix) $\mathbf{W}^* = \mathbf{y}\mathbf{x}^T$, and thus the recall strength $r = \mathbf{W} \cdot \mathbf{W}^*$ can be written as

$$
\begin{aligned}
\mathbf{W} \cdot \mathbf{W}^* &= \mathbf{W} \cdot (\mathbf{y}\mathbf{x}^T) \\
&= \text{Tr}[(\mathbf{y}\mathbf{x}^T)^T \mathbf{W}] \\
&= \text{Tr}[\mathbf{y}\mathbf{x}^T \mathbf{W}] \\
&= \mathbf{x} \cdot (\mathbf{y}^T \mathbf{W})^T \\
&= \mathbf{x}^T \mathbf{W}^T \mathbf{y} \\
&= (\mathbf{W}\mathbf{x})^T \mathbf{y} \\
&= \hat{\mathbf{y}} \cdot \mathbf{y},
\end{aligned}
\tag{20}
$$

corresponding to the accuracy of the prediction $\mathbf{y}$.

### Reinforcement learning

Let $\mathbf{x}$ be the input population activity representing state information, $\mathbf{W}$ be the output weights, and $\pi = \mathbf{W}\mathbf{x}$ be the output population activity representing unnormalized log probabilities of taking different actions $a \in \{1, ..., A\}$, so that $p(a) = \frac{e^{\beta \pi_a}}{\sum_j e^{\beta \pi_j}}$, where $\beta$ is a parameter controlling the stochasticity of the action selection. Let $\mathbf{a}$ be a one-hot vector indicating the sampled action, and $r = \pm 1$ be the scalar reward that results. For a reinforcement learning rule giving rise to a memory vector $\mathbf{W}^* = \text{reward} \cdot (\mathbf{a}\mathbf{x}^T)$, the recall strength $r = \mathbf{W} \cdot \mathbf{W}^*$ can be written as

$$
\begin{aligned}
\mathbf{W} \cdot \mathbf{W}^* &= \mathbf{W} \cdot \left( \text{reward} \cdot \mathbf{a}\mathbf{x}^T \right) \\
&= \text{reward} \cdot \left( \mathbf{a}\mathbf{x}^T \right)^T \mathbf{W}.
\end{aligned}
\tag{21}
$$

Following the same steps as the derivation for the supervised learning case, with $\mathbf{a}$ in place of $\mathbf{y}$, gives

$$
\begin{aligned}
\mathbf{W} \cdot \mathbf{W}^* &= \text{reward} \cdot \left( (\mathbf{W}\mathbf{x})^T \mathbf{a} \right) \\
&= \text{reward} \cdot \left( \pi^T \mathbf{a} \right) \\
&= \text{reward} \cdot \pi_a,
\end{aligned}
\tag{22}
$$

$$
= \text{reward} \cdot \left( \log p\left( a \right) + \log \left( \sum_j e^{\beta \pi_j} \right) \right),
\tag{23}
$$

corresponding to the unnormalized log probability with which the action $a$ was selected (which can be interpreted as the confidence in the selection), modulated by reward.

Computing this factor requires preserving the network's action probability distribution, extracting from it the probability of the sampled action, and multiplicatively scaling the result by the obtained reward.

## Autoassociative memory

Let $\mathbf{x}$ be the population activity and $\mathbf{W}$ be the recurrent weight matrix. For an autoassociative memory storage rule with memory vector $\mathbf{W}^* = \mathbf{xx}^T$, assuming that the weight matrix $\mathbf{W}$ can be approximated as a sum $\sum_i \mathbf{x}_i \mathbf{x}_i^T$ over prior plasticity-driven updates, then the recall strength $r = \mathbf{W} \cdot \mathbf{W}^*$ can be written as

$$
\begin{aligned}
\mathbf{W} \cdot \mathbf{W}^* &= \mathbf{W} \cdot (\mathbf{xx}^T) \\
&= \left( \sum_i (\mathbf{x}_i \mathbf{x}_i^T) \right) \cdot (\mathbf{xx}^T) \\
&= \left( \sum_i (\mathbf{x}_i \mathbf{x}_i^T) \cdot (\mathbf{xx}^T) \right) \\
&= \sum_i \mathrm{Tr}[(\mathbf{x}_i \mathbf{x}_i^T)^T (\mathbf{xx}^T)] \\
&= \sum_i \mathrm{Tr}[\mathbf{x}_i \mathbf{x}_i^T \mathbf{xx}^T] \\
&= \sum_i \mathbf{x}_i^T \mathbf{x} \mathrm{Tr}[\mathbf{x}_i \mathbf{x}^T] \\
&= \sum_i (\mathbf{x}_i^T \mathbf{x})^2,
\end{aligned}
\tag{24}
$$

corresponding the familiarity of the current pattern $\mathbf{x}$ relative to all previously seen patterns $\mathbf{x}_i$.

Familiarity also be computed with a separate familiarity readout trained alongside the recurrent weights. If the familiarity readout employs a Hebbian rule, the resulting estimate of familiarity will be equal to

$$
\sum_i \left( \mathbf{x}_i^T \mathbf{x} \right).
\tag{25}
$$

For uncorrelated patterns in a network below capacity, this strategy corresponds exactly to the true recall factor in the limit of large network size.

## Relationship between learnable timescale and capacity

We note that theoretical work on memory systems often focuses on memory *capacity*, the number of memories that can be reliably stored in the system (*Gardner, 1988*; *Fusi et al., 2005*; *Benna and Fusi, 2016*). Our learnable timescale metric is distinct from capacity. However, the two are closely linked in a particular regime. Suppose $P$ distinct reliable memories are reinforced independently at rates $\lambda_1, ..., \lambda_P$. In the regime in which the overall rate of reliable memory presentation $\lambda_{\mathrm{tot}} = \sum_i \lambda_i$ is small, the SNR of memory recall for memory $i$ will be the same as in the case of a single reliable memory with $\lambda = \lambda_i$ (*Figure 2—figure supplement 1*). Hence, for a fixed $\lambda_{\mathrm{tot}}$, and for simplicity assuming that distinct reliable memories are presented at equal rates $\lambda_i = \frac{1}{P}\lambda_{\mathrm{tot}}$ for all $i$, the learnable timescale $\tau^*$ of the system dictates its capacity, equal to $\tau^* \lambda_{\mathrm{tot}}$. We note that this correspondence does not hold in the case where most observed memories are reliable. In this work, however, we are interested primarily in the regime of scarce reliability, where recall-gated consolidation provides the most benefit. In this regime, we regard the learnable timescale as the most natural measure of system performance, as the primary obstacle to memory storage is the presence of long gaps between reinforcements of reliable memories.

## The effect of repeated reinforcement on memory dynamics without recall-gated consolidation

When memories can recur multiple times, the memory trace function $m(t)$ is no longer an adequate description of system behavior, as the synaptic updates from multiple presentations can combine. For the synaptic plasticity rules we consider here – the binary switch, the cascade model of Fusi et al., and the multivariable model of Benna & Fusi, this combination is approximately additive (*Benna and Fusi, 2016*). This is because for each of these plasicity rules, the change in distribution of synaptic states following the presentation of a memory is approximately independent of the existing synaptic state. The only dependencies are saturation effects – synapses which have reached the edge of their dynamic range – which can only lead to sub-additive behavior. Saturation effects can be avoided by making the dynamic range of synapses sufficiently large. Thus for these plasticity rules of interest we may consider additive memory trace combination to represent a close approximation (and a tight upper bound) on the combined memory trace strength.

For a reliable memory presented at times $t_1, ..., t_R$, and a population of synapses using additive plasticity rules, the current SNR at time $t$ can therefore be approximated as

$$\text{SNR}(t) = \sum_{i=1}^{R} m(t - t_i). \tag{26}$$

If memory presentations occur separated by regular intervals of length $\tau = \frac{1}{\lambda}$, we have

$$\text{SNR}(t) = \sum_{i=0}^{R-1} m(t - t_R + i\tau). \tag{27}$$

For the binary switch model, $m(t)$ decays exponentially with time constant $1/p$, and so the second term is negligible compared to the first. Hence the learnable timescale of the system is the same as the memory lifetime, approximately $1/p$. For target SNR threshold $\delta$, we require $p \geq \delta/\sqrt{N}$, so the best possible learnable timescale, optimizing over $p$, is $O(\sqrt{N}/\delta)$.

For the cascade model, $m(t)$ decays as $\frac{\sqrt{N}}{t \log T} e^{-t/T}$. For $t \gg T$ the exponential factor dominates, resulting in the same behavior as the binary switch model. For $t \ll T$, the exponential term approximately vanishes, so the following expression for $\text{SNR}(t)$ is a close approximation and tight upper bound:

$$
\begin{aligned}
\text{SNR}(t) \quad &\approx \frac{\sqrt{N}}{(t - t_R) \log T} + \int_1^{\min(R, T/\tau)} \frac{\sqrt{N}}{\tau \cdot t \log T} dt \\
&= \frac{\sqrt{N}}{(t - t_R) \log T} + \frac{\sqrt{N}}{\tau \log T} \log\left[\min(T/\tau, R)\right] \\
&= \frac{\sqrt{N}}{(t - t_R)(\log R + \log \tau)} + \frac{\sqrt{N}}{\tau(\log R + \log \tau)} \log R.
\end{aligned}
\tag{28}
$$

Again, for computing learnable timescale we are interested in when $t - t_R \approx \tau$, in which case:

$$\text{SNR}(t) \approx \frac{\sqrt{N}}{\tau} \frac{\log R}{\log R + \log \tau}. \tag{29}$$

For the multivariable model, $m(t)$ decays as $\sqrt{\frac{N}{t \log T}} e^{-t/T}$. Again we are primarily interested in the $t \ll T$ regime, in which the expression for $\text{SNR}(t)$ is approximately

$$
\begin{aligned}
\text{SNR}(t) \quad &\approx \frac{\sqrt{N}}{\sqrt{t - t_R}\sqrt{\log T}} + \int_1^{\min(R, T/\tau)} \frac{\sqrt{N}}{\sqrt{\tau \cdot t}\sqrt{\log T}} dt \\
&= \frac{\sqrt{N}}{\sqrt{t - t_R}\sqrt{\log T}} + \frac{2\sqrt{N}}{\sqrt{\tau}\sqrt{\log T}}\left(\sqrt{\log(T/\tau, R)} - 1\right).
\end{aligned}
\tag{30}
$$

This SNR is maximized for $T \approx R \cdot \tau$. And for computing learnable timescale we are interested in when $t - t_R \approx \tau$. So we have

$$\text{SNR}(t) = \frac{\sqrt{N}}{\sqrt{\tau \log(R \cdot \tau)}}\left(2\sqrt{R} - 1\right). \tag{31}$$

To compute the learnable timescale at target SNR $\delta$ for $1 \ll R \ll \tau$, we have $4NR \geq \tau \log(R\tau)\delta^2$, the solution of which is within logarithmic factors of $\mathcal{O}(RN)$.

The above calculations assume deterministic interarrival intervals of length $\tau$. In general, we are interested in an interarrival distribution $p(l)$ with mean $\tau$. However, we show numerically that for Weibull distributions with reasonable values of $k$ (not too close to zero), the true learnable timescale figures are very bounded very closely to our results above (*Figure 2—figure supplement 2*). Moreover, for the purpose of computing learnable timescale $\tau_\epsilon^\beta$ with error probability tolerance $\epsilon$, for sufficiently small $\epsilon$ the deterministic approximation represents an upper bound on the SNR for distributions with mean $\tau$. This is because to ensure high SNR with very high probability, deterministic intervals are a best case scenario, as stochastic interval lengths will with some nonzero probability deviate far above the mean.

## Bounds on an ideal synaptic consolidation model

In this section we show that no realistic synapse-local mechanism can achieve significantly better learnable timescale than $\mathcal{O}(RN)$, and hence that the ability of recall-gated systems consolidation to achieve learnable timescale scaling superlinearly with $R$ in some environments (see previous section) represents a qualitative advantage.

We consider a very general class of synaptic plasticity rules. In particular, we suppose a synapse can main tain a history of sequences of potentiation and depression events for arbitrarily long time windows and track the number of windows for which $\Delta$, the difference in number of potentiation and depression events, exceeds a threshold $\delta$. Let $p_{\text{reliable}}(\Delta; \tau)$ refer to the probability distribution of values of $\Delta$ after $\tau$ timesteps, given that a synapse is potentiated by the reliable memory of interest – and $p_{\text{unreliable}}$ refers to the analogous distribution for synapses subject only to potentiation by unreliable memories. After $\frac{T}{\tau}$ intervals of length $\tau$, for a synapse potentiated by the reliable memory, we have that

$$E\left[log\left(\frac{p\left(\text{reliable|data}\right)}{p\left(\text{unreliable|data}\right)}\right)\right] = \frac{T}{\tau}D_{\text{KL}}\left(p_{\text{reliable}}\left(\Delta; \tau\right) \| p_{\text{unreliable}}\left(\Delta; \tau\right)\right). \tag{32}$$

The memory can be considered retrievable with SNR of order $\mathcal{O}\left(1\right)$ once the expression above exceeds $\mathcal{O}\left(\frac{1}{N}\right)$ (since evidence can be accumulated across the $N$ synapses) for any choice of $\tau$ (since we are interested in the best achievable performance).

Now, for large enough $\tau$, $p_{\text{unreliable}}(\Delta; \tau)$ is approximately Gaussian with mean 0 and standard deviation $\sqrt{\tau}$. Conditioned on the reliable memory being presented $r$ times in $\tau$ timesteps, $p_{\text{reliable}}(\Delta)$ is approximately Gaussian with mean $r$ and standard deviation $\sqrt{\tau}$. The KL divergence between these two distributions is $\frac{r^2}{2\tau}$. Now consider the distribution $p_\tau(r)$ of number of repetitions $r$ that occur in a time window $\tau$. We want to find a value $r^{\text{max}}$ such that $P_\tau(r \geq r^{\text{max}}) \leq \frac{\tau}{T}$. From there we can assume that $r < r^{\text{max}}$ in any of the $\tau$-length intervals, since after $T$ timesteps we cannot reliably count on $r$ exceeding $r^{\text{max}}$ in any of the intervals.

For $\tau \leq M\left[I\right]$, the median of the distribution $p(l)$, note that $r^{\text{max}} \leq \log(T/\tau) \leq \log T$. For $\tau \leq c \cdot M\left[I\right]$, if $R > c\log T$ then at least one interval of less than length $M\left[I\right]$ contains at least $\log T$ repetitions. Hence $r^{\text{max}} \leq c\log T$. So conservatively we can take $r^{\text{max}} = \frac{\tau}{M[I]}\log T$.

Thus, our log probability expression above is bounded as follows

$$\begin{aligned} E\left[\log\left(\frac{p\left(\text{reliable|data}\right)}{p\left(\text{unreliable|data}\right)}\right)\right] &= \frac{T}{\tau}D_{\text{KL}}\left(p_{\text{reliable}}\left(\Delta; \tau\right) \| p_{\text{unreliable}}\left(\Delta; \tau\right)\right) \\ &\leq \frac{T}{\tau}\frac{r_{\text{max}^2}}{2\tau} \\ &= \frac{T}{\tau}\frac{\tau^2}{M\left[I\right]^2}\frac{\log^2\left(T\right)}{2\tau} \\ &= T\frac{\log^2\left(T\right)}{2M\left[I\right]^2}. \end{aligned}$$

Hence the KL divergence criterion becomes

$$T\frac{\log^2\left(T\right)}{2M\left[I\right]^2} \geq O\left(\frac{1}{N}\right), \tag{33}$$

or equivalently,

$$TN\log^2 T \geq 2M\left[I\right]^2. \tag{34}$$

The number of repetitions is $R \approx T/E\left[I\right]$, giving

$$RN\log^2\left(RE\left[I\right]\right) \geq \frac{2M\left[I\right]^2}{E\left[I\right]}, \tag{35}$$

$$RN \log^2 (RE\,[I]) \geq 2E\,[I]\,, \tag{36}$$

assuming $M\,[I] \sim O\,(E\,[I])$. For the interarrival distributions we consider (of the Weibull family), $M\,[I] < E\,[I]$ so this is a conservative assumption. Hence the learnable timescale of any population using only synapse-local plasticity rules is no greater than the solution for $E\,[I]$ of the equation above. We have

$$RN \geq 2E\,[I]\,/\log^2 (RE\,[I])\,, \tag{37}$$

the solution of which is within logarithmic factors of $\mathcal{O}\,(RN)$.

## Scaling behavior of the STM/LTM model

In the recall-gated consolidation model, the overlap $r = \mathbf{w}_{\mathrm{STM}} \cdot \mathbf{w}_{\mathrm{STM}}^*$ indicates the recall strength of memory $x$ given the current synaptic state of the STM. LTM plasticity is modulated by a factor $g(r)$ – we refer to $g$ as the "gating function" and $r$ as the STM recall strength. We assume for now that the gating function $g(r)$ is chosen to be a threshold function, $g\,(r) = H\,(r - \theta)$, where $r$ is the SNR of the memory overlap, $H$ is the Heaviside step function, and $\theta$ is referred to as the "consolidation threshold." With this choice, unreliable memories will be consolidated at a rate of $1 - \Phi\,(\theta)$, where $\Phi$ is the CDF of the normal distribution, in the limit of large system size $N$.

Suppose a memory $\mathbf{w}^*$ is presented twice with interval $I$. Then the SNR at the second presentation will be lower-bounded by $m(I)$, in expectation. It follows that the rate at which reliable memories will be consolidated at for the gating function above is lower bounded by $P(I < m^{-1}(\theta))$. After $R$ repetitions of the reliable memory, the probability that consolidation has occurred will be at least

$$1 - (1 - P(I < m^{-1}(\theta)))^R. \tag{38}$$

We are interested in the maximum $\theta$ for which this expression exceeds $1 - \epsilon$ – this is the most stringent consolidation threshold we can set while still ensuring consolidation of the reliable memory with high probability. This value of $\theta$ is given by

$$R \log(1 - P(I < m^{-1}(\theta)) = \log \epsilon \tag{39}$$

If $R$ is large then the solution will be such that $P(I < m^{-1}(\theta))$ is small, enabling the approximation:

$$P\left(I < m^{-1}\,(\theta)\right) = -\log \epsilon/R \tag{40}$$

For tractability we consider, as our family of interarrival distributions, Weibull distributions with regularity parameter $k$. The cumulative distribution function is

$$P\,(I \leq t) = 1 - e^{\left(t\Gamma\,(1+1/k)/\tau\right)^k}. \tag{41}$$

For $t << \tau$, this is approximated as

$$P\,(I \leq t) \approx \left(t\Gamma\,(1 + 1/k)\,/\tau\right)^k. \tag{42}$$

Importantly, $P\,(I \leq t)$ decays as $t^k$. Thus, increasing the number of repetitions $R$ has the effect of scaling the $\tau$ that satisfies *Equation 40* by $R^{1/k}$. That is, for a fixed $\theta$, and hence a fixed degree of amplification $\tau_{\mathrm{LTM}}/\tau$ of the effective rate of reliable memories in the LTM, the maximum $\tau$ achieving that SNR with probability $1 - \epsilon$ (i.e. the learnable timescale $\tau_\beta^\epsilon$) scales as $\mathcal{O}\left(R^{1/k}\right)$.

For a gating function threshold $\theta$, the corresponding SNR in the LTM will be the SNR induced by an interarrival distribution $I_{\mathrm{LTM}}$ with mean value

$$E\,[I_{\mathrm{LTM}}] = \frac{P\left(I < m^{-1}\,(\theta)\right)}{1 - \Phi\,(\theta)} E\,[I]\,. \tag{43}$$

Since $1 - \Phi\,(\theta)$ decays much more rapidly than any power of $m^{-1}\,(\theta)$, it follows that $E\,[I_{\mathrm{LTM}}]$ can be made $O(1)$, and thus the SNR of the LTM can become $O\left(\sqrt{N}\right)$, for relatively small values of $\theta$ (and hence a small number of required repetitions). In other words, for a fixed number of expected

memory repetitions, the learnable timescale of the LTM decreases only slightly as the target SNR is raised from $O(1)$ to $O\left(\sqrt{N}\right)$.

Note that if $P$ different reliable memories are present, then $E\left[I_{\text{LTM}}\right]$ for any given reliable memory will be lower-bounded by $\mathcal{O}\left(P\right)$ instead of $\mathcal{O}\left(1\right)$. The induced SNR for any given reliable memory in the LTM will in this case be of order $m(P)$, rather than $O\left(\sqrt{N}\right)$.

## Optimal sparsity calculations

We now consider the case of sparse memories – those which potentiate a fraction $f$ of synapses. Consider the behavior of a single population of binary synapses employing the binary switch plasticity rule. We modify the plasticity rule slightly so that potentiation flips the state of a synapse with probability $p$ and depression with probability $\frac{f}{1-f}p$, to ensure that the fractions of potentiated and depressed synapses remain balanced.

We consider an environment with a single reliable memory that is presented with probability $\lambda$ at each time step (otherwise, a randomly sampled unreliable memory is presented). We can compute the behavior analytically by tracking how the distributions of $u$ (the output neuron response to true stimuli) and $v$ (the output neuron response to noise) evolve over time. We assume that the coding level $f$ is sufficiently small that terms of order $O\left(f^2\right)$ may be ignored.

Due to the balanced plasticity rule, $\frac{1}{2}$ of synapses are strong at any given time, so the mean response $v^*$ to a randomly sampled noise pattern is $\frac{1}{2}$. The variance of $v$ is also constant and equal to $\frac{1}{4Nf}$.

The evolution of $u$ is a stochastic process that, in the limit of large $Nf$ (i.e. a large number of active neurons for each stimulus), can be described as an Ornstein-Uhlenbeck (OU) process:

$$\Delta u = \theta\left(u^* - u\right) + \epsilon. \tag{44}$$

where $\epsilon \sim N(0, \sigma^2)$

In the limit of small $f$ we have:

$$\theta = p\left(2f + \lambda\right) \tag{45}$$

$$u^* = \frac{f + \lambda}{2f + \lambda} = \frac{1}{2} + \frac{\lambda}{4f + 2\lambda}. \tag{46}$$

The quantity $u^*$ determines the asymptotic mean of $u$ and the quantity $\theta$ determines the rate at which $u$ converges to this mean. Immediately we see that $u^*$ scales with the frequency $\lambda$ with which the true stimulus is presented, and that the rate of convergence (speed of learning) is proportional to $p$.

By well-known properties of OU processes, the asymptotic variance of $u$ is equal to $\frac{\sigma^2}{2\theta}$. In the small-$p$ limit, this quantity comes out to

$$\text{Var}\left(u\right) = \frac{1}{4Nf}. \tag{47}$$

Note that in the low-$p$ limit (slow learning rate) this is the same as the variance of $v$. Thus in the limit of slow learning, we have that

$$E\left[u - v\right] = \frac{\lambda}{4f + 2\lambda}, \tag{48}$$

$$\text{Var}\left(u - v\right) = \frac{1}{2Nf}. \tag{49}$$

And thus

$$\text{SNR} = \frac{N}{2}\frac{f\lambda^2}{\left(2f + \lambda\right)^2}. \tag{50}$$

From this expression we can see that for a given $f$, the asymptotic SNR always increases with $\lambda$ and $N$. For a given $\lambda$, we would like to maximize this expression with respect to $f$.

$$\nabla_f \text{SNR} = \frac{N}{2} \frac{(2f + \lambda)^2 \lambda^2 - f\lambda^2 \cdot 4(2f + \lambda)}{(2f + \lambda)^4}.$$
(51)

This expression equals zero when

$$(2f + \lambda)^2 \lambda^2 - f\lambda^2 \cdot 4(2f + \lambda) = 0$$
(52)

$$(2f + \lambda) = 4f$$
(53)

$$f = \frac{1}{2}\lambda,$$
(54)

so the asymptotic SNR is maximized for $f = \frac{1}{2}\lambda$. That is, the optimal coding level is proportional to the frequency with which reliable (as opposed to unreliable) stimuli are observed in the environment.

