## [Editor Report · eLife assessment]

This **fundamental** work proposes a novel mechanism for memory consolidation where short-term memory provides a gating signal for memories to be consolidated into long-term storage. The work combines extensive analytical and numerical work applied to three different scenarios and provides a **convincing** analysis of the benefits of the proposed model, although some of the analyses are limited to the type of memory consolidation the authors consider (and don't consider), which limits the impact. The work will be of interest to neuroscientists and many other researchers interested in the mechanistic underpinnings of memory.

---

## [Referee Report · Reviewer #2 (Public Review)]

Summary:

In the manuscript the authors suggest a computational mechanism called recall-gated consolidation, which prioritizes the storage of previously experienced synaptic updates in memory. The authors investigate the mechanism with different types of learning problems including supervised learning, reinforcement learning, and unsupervised auto-associative memory. They rigorously analyse the general mechanism and provide valuable insights into its benefits.

Strengths:

The authors establish a general theoretical framework, which they translate into three concrete learning problems. For each, they define an individual mathematical formulation. Finally, they extensively analyse the suggested mechanism in terms of memory recall, consolidation dynamics, and learnable timescales.

The presented model of recall-gated consolidation covers various aspects of synaptic plasticity, memory recall, and the influence of gating functions on memory storage and retrieval. The model's predictions align with observed spaced learning effects.

The authors conduct simulations to validate the recall-gated consolidation model's predictions, and their simulated results align with theoretical predictions. These simulations demonstrate the model's advantages over consolidating any memory and showcase its potential application to various learning tasks.

The suggestion of a novel consolidation mechanism provides a good starting point to investigate memory consolidation in diverse neural systems and may inspire artificial learning algorithms.

Weaknesses:

I appreciate that the authors devoted a specific section to the model's predictions, and point out how the model connects to experimental findings in various model organisms. However, the connection is rather weak and the model needs to make more specific predictions to be distinguishable from other theories of memory consolidation (e.g. those that the authors discuss) and verifiable by experimental data.

The model is not compared to other consolidation models in terms of performance and how much it increases the signal-to-noise ratio. It is only compared to a simple STM or a parallel LTM, which I understand to be essentially the same as the STM but with a different timescale (so not really an alternative consolidation model). It would be nice to compare the model to an actual or more sophisticated existing consolidation model to allow for a fairer comparison.

The article is lengthy and dense and it could be clearer. Some sections are highly technical and may be challenging to follow. It could benefit from more concise summaries and visual aids to help convey key points.

---

## [Referee Report · Reviewer #3 (Public Review)]

Summary:

In their article Jack Lindsey and Ashok Litwin-Kumar describe a new model for systems memory consolidation. Their idea is that a short-term memory acts not as a teacher for a long-term memory - as is common in most complementary learning systems -, but as a selection module that determines which memories are eligible for long term storage. The criterion for the consolidation of a given memory is a sufficient strength of recall in the short term memory.

The authors provide an in-depth analysis of the suggested mechanism. They demonstrate that it allows substantially higher SNRs than previous synaptic consolidation models, provide an extensive mathematical treatment of the suggested mechanism, show that the required recall strength can be computed in a biologically plausible way for three different learning paradigms, and illustrate how the mechanism can explain spaced training effects.

Strengths:

The suggested consolidation mechanism is novel and provides a very interesting alternative to the classical view of complementary learning systems. The analysis is thorough and convincing.

Weaknesses:

The main weakness of the paper is the equation of recall strength with the synaptic changes brought about by the presentation of a stimulus. In most models of learning, synaptic changes are driven by an error signal and hence cease once the task has been learned. The suggested consolidation mechanism would stop at that point, although recall is still fine. The authors should discuss other notions of recall strength that would allow memory consolidation to continue after the initial learning phase. Aside from that, I have only a few technical comments that I'm sure the authors can address with a reasonable amount of work.

---

## [Author Response]

The following is the authors’ response to the original reviews.

In light of some reviewer comments requesting more clarity on the relationship between our model and prior theoretical studies of systems consolidation, we propose a modification to the title of our manuscript: “Selective consolidation of learning and memory via recall-gated plasticity.” We believe this title better reflects the key distinguishing feature of our model, that it *selectively* consolidates only a subset of memories, and also highlights the model’s applicability to task learning as well as memory storage.

Major comments:

Reviewer #3’s primary concern with the paper is the following: “The main weakness of the paper is the equation of recall strength with the synaptic changes brought about by the presentation of a stimulus. In most models of learning, synaptic changes are driven by an error signal and hence cease once the task has been learned. The suggested consolidation mechanism would stop at that point, although recall is still fine. The authors should discuss other notions of recall strength that would allow memory consolidation to continue after the initial learning phase.”

We thank the reviewer for drawing attention to this issue, which primarily results from a poor that memories should be interpreted as actual synaptic weight updates, ∆𝑤 and thus in the context choice of notation on our part. Our decision to denote memories as gives the impression of supervised learning would go to zero when the task is learned. However, in the formalism of our model, memories are in fact better interpreted as *target* values of synaptic weights, and the synaptic model/plasticity rule is responsible for converting these target values into synaptic weight updates. We were unclear on this point in our initial submission, because our paper primarily considers binary synaptic weights, where target synaptic weights have a one-to-one correspondence with candidate synaptic weight updates. We have updated the paper to use w* to refer to memories, which we hope resolves this confusion, and have updated our introduction to the term “memory” to reflect their interpretation as target synaptic weight values. We have also updated the paper’s language to more clearly disambiguate between the “learning rule,” which determines how the memory vector (target synaptic weight vectors) are derived from task variables, and the “plasticity rule,” which governs how these are translated into actual synaptic weight updates. We acknowledge that our manuscript still does not explicitly consider a plasticity rule that is sensitive to continuous error error signals, as our analysis is restricted to binary weights. However, we believe that the updated notation and exposition makes it more clear that our model could be applied in such a case.

Reviewer #1 brought up that our framework cannot capture “single-shot learning, for example, under fear conditioning or if a presented stimulus is astonishing.” Reviewer #2 raised a related question of how our model “relates to the opposite more intuitive idea, that novel surprising experiences should be stored in memory, as the familiar ones are presumably already stored.”

We agree that the built-in inability to consolidate memories after a single experience is a limitation of our model, and that extreme novelty is one factor (among others, such as salience or reward) that might incentivize one-shot consolidation. We have added a comment to the discussion to acknowledge these points (added text in bold): “ Moreover, in real neural circuits, additional factors besides recall, such as reward or salience, are likely to influence consolidation as well. For instance, a sufficiently salient event should be stored in long-term memory even if encountered only once. Furthermore, while in our model familiarity drives consolidation, certain forms of novelty may also incentivize consolidation, raising the prospect of a non-monotonic relationship between consolidation probability and familiarity.” We agree that future work should address the combined influence of recall (as in our model) and other factors on the propensity to consolidate a memory.

Reviewer #1 requested, “a comparison/discussion of the wide range of models on synaptic tagging for consolidation by various types of signals. Notably, studies from Wulfram Gerstner's group (e.g., Brea, J., Clayton, N. S., & Gerstner, W. (2023). Computational models of episodic-like memory in food-caching birds. Nature Communications, 14(1); and studies on surprise).”

We thank the reviewer for the reference, which we have added to the manuscript. The model of Brea et al.(2023) is similar to that of Roxin & Fusi (2013), in that consolidation consists of “copying” synaptic weights from one population to another. As a result, just like the model of Roxin & Fusi (2013), this model does not provide the benefit that our model offers in the context of consolidating repeatedly recurring memories. However, the model of Brea et al. does have other interesting properties – for instance, it affords the ability to decode the age of a memory, which our model does not. We have added a comment on this point in the subsection of the Discussion tilted “Other models of systems consolidation.”

Reviewer #2 noted, “While the article extensively discusses the strengths and advantages of the recall-gated consolidation model, it provides a limited discussion of potential limitations or shortcomings of the model, such as the missing feature of generalization, which is part of previous consolidation models. The model is not compared to other consolidation models in terms of performance and how much it increases the signal-to-noise ratio.”

We agree that our work does not consider the notion of generalization and associated changes to representational geometry that accompany consolidation, which is the focus of many other studies on consolidation. We have further highlighted this limitation in the discussion. Regarding the comparison to other models, this is a tricky point as the desiderata we emphasize in this study (the ability to recall memories that are intermittently reinforced) is not the focus of other studies. Indeed, our focus is primarily on the ability of systems consolidation to be *selective* in which memories are consolidated, which is somewhat orthogonal to the focus of many other theoretical studies of consolidation. We have updated some wording in the introduction to emphasize this focus.

Additional comments made by reviewer #1

Reviewer #1 pointed out issues in the clarity of Fig. 2A. We have added substantial clarifying text to the figure caption.Reviewer #1 pointed out lack of clarity in our introduction to the terms “reliability” and “reinforcement.” We have now made it more clear what we mean by these terms the first time they are used.

We have updated our definition of “recall” to use the term “recall factor,” which is how we refer to it subsequently in the paper.

We have made explicit in the main text our simplifying assumption that memories are mean-centered.

We have made consistent our use of “forgetting curve” and “memory trace”.

Additional comments made by reviewer #2

We have added a comment in the discussion acknowledging alternative interpretations of the result of Terada et al. (2021)

We have significantly expanded the discussion of findings about the mushroom body to make it accessible to readers who do not specialize in this area. We hope this clarifies the nature of the experimental finding, which uncovered a circuit that performs a strikingly clean implementation of our model.

The reviewer expresses concern that the songbird study (Tachibana et al., 2022) does not provide direct evidence for consolidation being gated by familiarity of patterns of activity. Indeed, the experimental finding is one-step removed from the direct predictions of our model. That said, the finding – that the *rate* of consolidation increases with performance – is highly nontrivial, and is predicted by our model when applied to reinforcement learning tasks. We have added a comment to the discussion acknowledging that this experimental support for our model is behavioral and not mechanistic.

We do not regard it as completely trivial that the parallel LTM model performs roughly the same as the STM model, since a slower learning rate can achieve a higher SNR (as in Fig. 2C). Nevertheless we have added wording to the main text around Fig. 4B to note that the result is not too surprising.

We have added a sentence that clarifies the goal / question of our paper earlier on in the introduction.

We have updated Figure 3 by labeling the key components of the schematics and adding more detail to the legend, as suggested by the reviewer. We also reordered the figure panels as suggested.

Additional comments made by reviewer #3:

We have clarified in the main text that Fig. 2C and all results from Fig. 4 onward are derived from an ideal observer model (which we also more clearly define).

We have now emphasized in the main text that the derivations of the recall factors for specific learning rules are derived in the Supplementary Information.

We have highlighted more clearly in the main text that the recall factors associated with specific learning rules may correspond to other notions that do not intuitively correspond to “recall,” and have added a pointer to Fig. 3A where these interpretations are spelled out.

We have added references corresponding to the types of learning rules we consider.

The cutoffs / piecewise-looking behavior of plots in Fig. 4 are primarily the result of finite N, which limits the maximum SNR of the system, rather than coarse sampling of parameter values.

Thank you for pointing out the error in the legend in Fig. 5D (also affected Supp Fig. S7/S8), which is now fixed.

The reference to the nonexistence panel Fig. 5G has been removed.

As the reviewer points out, the use of a binary action output in our reinforcement learning task renders it quite similar to the supervised learning task, making the example less compelling. In the revised manuscript we have updated the RL simulation to use three actions. Note also that in our original submission the network outputs represented action probabilities directly (which is straightforward to do for binary actions, but not for more than two available actions). In order to parameterize a policy when more than two actions are available, we sample actions using a softmax policy, as is more standard in the field and as the reviewer suggested. The associated recall factor is still a product of reward and a “confidence factor,” and the confidence factor is still the value of the network output in the unit corresponding to the chosen action, but in the updated implementation this factor is equal to log⁡p(a)+log⁡∑jeπj, similar (though with a sign difference) to the reviewer’s suggestion. We believe these updates make our RL implementation and simulation more compelling, as it allows them to be applied to tasks with arbitrary numbers of actions.

Additional minor comments

The reviewers made a number of other specific line-by-line wording suggestions, typo corrections,